# Transcriptome data reveal conserved patterns of fruiting body development and response to heat stress in the mushroom-forming fungus *Flammulina filiformis*

Xiao-Bin Liu[1,2], En-Hua Xia[3], Meng Li[4], Yang-Yang Cui[1,2], Pan-Meng Wang[1,2], Jin-Xia Zhang[5,6], Bao-Gui Xie[7], Jian-Ping Xu[8], Jun-Jie Yan[7], Jing Li[1,2,9], László G. Nagy[10], Zhu L. Yang[1,2]*

**1** CAS Key Laboratory for Plant Diversity and Biogeography of East Asia, Kunming Institute of Botany, Chinese Academy of Sciences, Kunming, Yunnan, China, **2** Yunnan Key Laboratory for Fungal Diversity and Green Development, Kunming, Yunnan, China, **3** State Key Laboratory of Tea Plant Biology and Utilization, Anhui Agricultural University, Hefei, Anhui, China, **4** Yunnan Tobacco Science Research Institute, Kunming, China, **5** Institute of Agricultural Resources and Regional Planning, Chinese Academy of Agricultural Sciences, Beijing, China, **6** Key Laboratory of Microbial Resources, Ministry of Agriculture and Rural Affairs, Beijing, China, **7** Mycological Research Center, College of Life Sciences, Fujian Agriculture and Forestry University, Fuzhou, China, **8** Department of Biology, McMaster University, Hamilton, Ontario, Canada, **9** Key Laboratory of Conservation and Utilization for Bioresources and Key Laboratory of Microbial Diversity in Southwest China, Ministry of Education, Yunnan University, Kunming, Yunnan, China, **10** Synthetic and Systems Biology Unit, Institute of Biochemistry, Biological Research Centre, Szeged, Szeged, Hungary

* fungi@mail.kib.ac.cn

**Data Availability Statement:** All raw reads were deposited in NCBI Sequence Read Archive (SRA,

## Abstract

Mushroom-forming fungi are complex multicellular organisms that form the basis of a large industry, yet, our understanding of the mechanisms of mushroom development and its responses to various stresses remains limited. The winter mushroom (*Flammulina filiformis*) is cultivated at a large commercial scale in East Asia and is a species with a preference for low temperatures. This study investigated fruiting body development in *F. filiformis* by comparing transcriptomes of 4 developmental stages, and compared the developmental genes to a 200-genome dataset to identify conserved genes involved in fruiting body development, and examined the response of heat sensitive and -resistant strains to heat stress. Our data revealed widely conserved genes involved in primordium development of *F. filiformis*, many of which originated before the emergence of the Agaricomycetes, indicating co-option for complex multicellularity during evolution. We also revealed several notable fruiting-specific genes, including the genes with conserved stipe-specific expression patterns and the others which related to sexual development, water absorption, basidium formation and sporulation, among others. Comparative analysis revealed that heat stress induced more genes in the heat resistant strain (M1) than in the heat sensitive one (XR). Of particular importance are the *hsp70*, *hsp90* and *fes1* genes, which may facilitate the adjustment to heat stress in the early stages of fruiting body development. These data highlighted novel genes involved in complex multicellular development in fungi and aid further studies on gene function and efforts to improve the productivity and heat tolerance in mushroom-forming fungi.

http://www.ncbi.nlm.nih.gov/Traces/sra) with accession number of PRJNA557510.

**Funding:** This study was supported by the Strategic Priority Research Program of Chinese Academy of Sciences (No. XDB31000000); Yunnan Ten-Thousand-Talents Plan - Yunling Scholar Project; and the National Basic Research Program of China (973 Program, No. 2014CB138305). LGN acknowledges support from the European Research Council (grant no. 758161 to L.G.N.) and the National Research, Development and Innovation office (Contract No. Ginop-2.3.2-15-00001, to LGN).

## Introduction

Mushroom-forming fungi are widely distributed through Earth's ecosystems. They play essential roles in nutrient cycling, environmental protection, plant and animal health [1–3]. Mushrooms are also important food sources and produce molecules with therapeutic activities and enzymes that can be applied in bioconversion [4, 5]. Furthermore, the fruiting body is a complex multicellular structure [6], whose complexity level resembles that of multicellular plants and animals. Therefore, understanding fruiting body development is important also from the perspective of understanding major evolutionary transitions.

Fruiting body development starts with the aggregation of aerial dikaryotic hyphae under suitable environmental conditions (nutrient, light, and temperature etc.) [7–13]. These aggregates continuously develop into primordia, which further differentiate into mature fruiting bodies [7–13]. Then, karyogamy and meiosis take place in the basidia within the hymenium of the fruiting body, and additional mitosis results in basidiospores [7–13]. *Coprinopsis cinerea* and *Schizophyllum commune* were used as the main model species to study the mechanisms of mushroom formation, due to their short life cycles and suitability for genetic manipulation [7–9]. Studies on these two model species using tools such as DNA-mediated transformation, RNA interference, and CRISPR/Cas9 etc. have pioneered our knowledge of the multicellular development, mating pheromone, and receptor signaling pathways in the Agaricomycetes [14–18]. More recently, studies also focused on ecologically or economically important non-model species, which included the saprotrophic fungi (*Agaricus bisporus*, *Flammulina filiformis*, *Lentinula edodes*, *Lentinus tigrinus*, *Cyclocybe aegerita*), plant pathogen (*Armillaria ostoyae*), and the ectomycorrhizal fungi (*Laccaria bicolor*) [2, 19–26]. These studies broadened our knowledge on fruiting body development and also highlighted conserved expression patterns of some key developmental genes (such as the genes encoding light receptors (white collar complex), transcription factors (*c2h2*, *hom1*, *hom2*), CAZyme and F-box protein etc.), indicating conserved molecular mechanisms in multicellular complexity in Agaricomycetes. However, mushroom development is a highly organized process, and genetic drivers of spatial and temporal differentiation events are not known, and our understanding of mushroom formation in other ecologically and economically important mushroom-forming fungi is still in its infancy [12, 13, 27, 28].

The winter mushroom or enokitake, *Flammulina filiformis* (formerly known as *F. velutipes*) [29], is cultivated at large scales in East Asia [30–32]. A comprehensive understanding of fruiting body development of this mushroom would not only benefit its production, but can also help to uncover conserved molecular mechanisms of development in the Agaricomycetes. Commercial scale cultivation of this mushroom requires a low temperature ($\leq 15°C$) (since the wild strain commonly fruiting during late autumn to early spring), which costs large amounts of energy, especially during summer in East Asia [29, 33, 34]. Fortunately, a heat resistant strain (called M1 after Mingjin1) has been isolated in subtropical areas (Fujian province) of China in summer that can fruit at 23°C and thus has great potentials in strain improvement and should be subject to studies of heat resistance [33].

Previous studies revealed molecular details in fruiting body development in *Flammulina* species at the transcriptome, proteome, single gene or protein level [20, 35–39]. A series of genes associated with mushroom formation, including the mating type genes, hydrophobins, and fruiting body specific genes have been identified [20, 31, 40]. Researches also investigated at least two genes controlling fruiting at >15°C based on hybridization analysis [34]. However, the molecular response to heat stress, in particular those of the well-known heat shock protein coding gene *hsp70*, *hsp90* and other molecular chaperons etc., in *F. filiformis* remains unknown. Although previous studies provided us with a basic understanding of fruiting body

development of *F. filiformis* or its closely related *F. velutipes*, our knowledge about the fruiting body development and heat stress response of this mushroom is still incomplete.

In this study, we aimed to (i) uncover key fruiting body genes in various developmental stages; (ii) investigate whether conserved developmental patterns exist in *F. filiformis* and other Agaricomycetes; and (iii) understand the responses to heat stress of heat sensitive (XR) and resistant (M1) strains and identify the key heat stress response genes in *F. filiformis*. We sampled RNA in different developmental stages of the M1 strain, and from the primordium stage grown at 10˚C and 18˚C of both the M1 and XR strains, for studying fruiting body development and its response to heat stress, respectively. Our results identified conserved gene expression patterns of fruiting body development in the Agaricomycetes and revealed that the heat tolerant strain M1 differentially expressed more genes in response to heat stress than the heat sensitive strain XR.

## Materials and methods

### Strains and culture conditions

The heat tolerant strain M1 (CGMCC5.2219) was domesticated from a wild strain collected in subtropical areas in China in summer (Fujian province). The heat sensitive strain XR (CGMCC5.2218) was isolated from a mushroom market, this strain was imported from Japan [33]. Both of them are deposit in the Chinese General Microbiological Culture Collection Center (CGMCC). They were grown on enriched Potato Dextrose Agar (PDA) medium (0.05% $KH_2PO_4$, 0.05% $MgSO_4$, 2% glucose, 0.2% yeast extract, 0.2% peptone, 1.8% agar) in 90 mm Petri dish for 10 days at 23 ˚C. Then, the mycelium was inoculated in liquid cultures in 500ml Erlenmeyer flask containing 200ml enriched Potato Dextrose Broth (PDB) medium (0.05% $KH_2PO_4$, 0.05% $MgSO_4$, 2% glucose, 0.2% yeast extract, 0.2% peptone), shaken at 150 r.p.m for 10 days at 23 ˚C. Afterwards, the mycelium was inoculated to a growth medium consisting of 90% cottonseed hull, 10% wheat bran, and 65% water in 1100 ml disposable bottles after sterilization. Inoculated bottles were incubated at 23 ˚C under dark conditions for 30 days, and then the mycelium scratched to emulates disturbance and transferred to 23 ˚C with 95% humidity for fruiting.

### Sample collection for RNA-seq

We selected different developmental stages of strain M1 grown at 10 ˚C (except for the mycelium, which was grown at 23 ˚C). We collected the vegetative mycelium (VM), the primordium (P10-M1), the young fruiting body cap (YFBC), the fruiting body cap (FBC), the young fruiting body stipe (YFBS), and the fruiting body stipe (FBS) (Fig 1A). For the heat stress response study, we collected the primordia, young fruiting body, and fruiting body of M1 strain grown at 18 ˚C (P18-M1), as well as the primordia, young fruiting body and fruiting body of XR strain grown at 10 ˚C and 18 ˚C respectively (Fig 1B).

### Total RNA preparation and transcriptome sequencing

Samples from three biological replicates were flash-frozen in liquid nitrogen and stored at -80 ˚C. Total RNA of each sample was extracted using the RNAprep Pure toolkit, following the manufacturer's protocol (TIANGEN, Beijing, China). Sequencing libraries were generated using NEBNext Ultra RNA Library Prep Kit for Illumina (NEB, USA) following the manufacturer's recommendations and index codes were added to attribute sequences to each sample. The clustering of the index-coded samples was performed on a cBot Cluster Generation System using TruSeq PE Cluster Kit v3-cBot-HS (Illumina) according to the manufacturer's

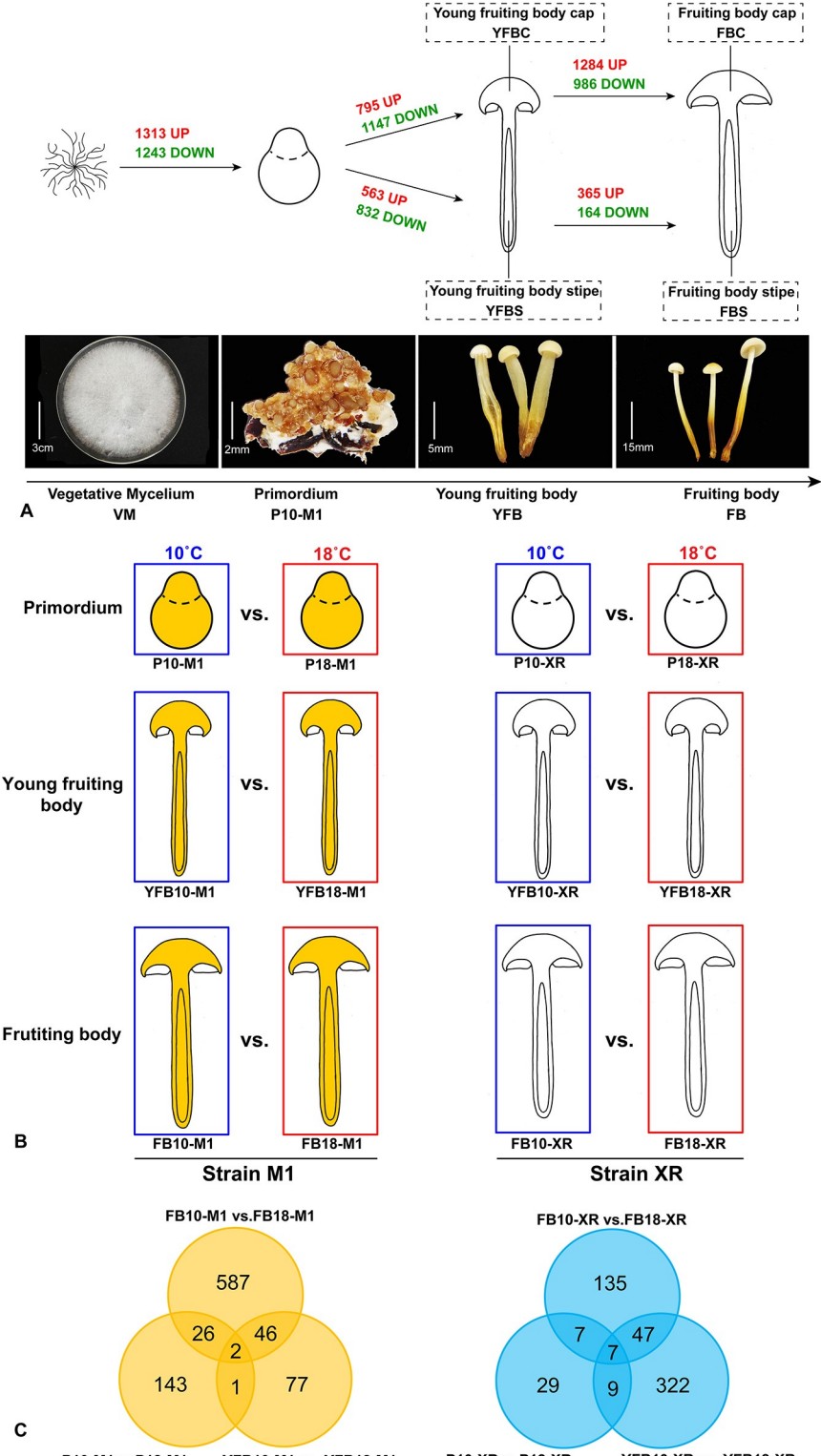

**Fig 1. Overview of the sampling and differential expression analyses used in this study.** A: fruiting body development. B: heat stress; C: Venn diagram of the numbers of up-regulated genes at 18°C compared to 10°C in each developmental stage of the M1 and XR strain.

instructions. After cluster generation, the library preparations were sequenced on an Illumina Hiseq platform and 125 bp/150 bp paired-end reads were generated. All raw-sequence reads data were deposited in NCBI Sequence Read Archive (SRA, http://www.ncbi.nlm.nih.gov/Traces/sra) with accession number of PRJNA557510.

### Read mapping to the reference genome, FPKM and gene annotation

Clean data were obtained by removing reads containing adapters, reads containing poly-N and low-quality reads from raw data through Trimmomatic (v.0.33) [41]. Then, we used HISAT2 (v.2.10) [42] to map the clean reads to the reference genome which assembled to chromosome level [20], and employed StringTie (v.1.3.4) [43] to calculate each gene's FPKM value. All of the genes were annotated using local BLASTX programs against the Nr, SwissPort, GO and PFAM databases.

### CAZyme gene annotation

Carbohydrate-active enzymes (CAZymes) were classified separately by HMM search of dbCAN HMMs 4.0 [44] (default cutoff threshold) and BLASTP search of the CAZy database [45] (evalue $< = 1x10^{-6}$ and coverage $> = 0.2$, maximum hit number is 500).

### Differential gene expression analysis

We performed differential gene expression analyses on each adjacent developmental stage of the M1 strain (Fig 1A) as well as the primordium stage grown at 10 ˚C vs 18 ˚C of the M1 and XR strains (Fig 1B). Analyses were based on three biological replicates per condition and were performed using the DESeq package (1.18.0) [46]. The input data of young fruiting body grown at 10 ˚C of M1 (YFB10-M1) were used as the pooled reads of young fruiting body cap (YFBC) and young fruiting body stipe (YFBS) of M1, and the input data of fruiting body grown at 10˚C of M1 (FB10-M1) were used as the pooled reads of fruiting body cap (FBC) and fruiting stipe (FBS) of M1. Genes with log2 (fold change) $\geq 1$ and Padj $\leq 0.05$ were considered as differentially expressed gene (DEG).

### Gene network construction and visualization

Co-expression networks were constructed using the WGCNA package in R [47]. Genes with averaged FPKM from three replicates higher than 1 in at least one sample were input to WGCNA unsigned co-expression network analysis (S1 Table). The modules were obtained using the step-by-step network construction function on block-wise modules with default settings, except that the power is 24 for fruiting body development analysis, 20 for M1 and 10 for XR in heat stress response analysis. TOM-Type was set to signed, minModuleSize to 30, and mergeCutHeight to 0.25. The networks were visualized using Cytoscape (v.3.5.1) [48].

### Comparative genomic analysis

In order to check the conservation level of the developmentally regulated genes? during fruiting body development, analyzed a 201 genome dataset (ranging from unicellular yeasts to filamentous and complex multicellular fungi which also included *F. filiformis* in this study) and the corresponding phylogenetic tree taken from a previous study [13]. Conservation of genes was assessed based on the phylogeny, by assessing the presence/absence of genes across the panel of species.

## qRT-PCR analysis

Reverse transcription of RNA (1ug) in a 20 μL reaction volume was performed using TURE-script 1st Stand cDNA SYNTHESIS Kit, following the manufacturer's protocol (Aidlab, Beijing, China). Reactions were incubated at 42 ˚C for 60min, and 65 ˚C for 10min. The amplifications were performed using 5 μL SYBR qPCR Mix, 0.5 μL forward primer, 0.5 μL reverse primer and 1 μL cDNA, and 3 μL ddH2O in a final volume of 10 μL. The cycling parameters were 95 ˚C for 3 min followed by 30 cycles of 95 ˚C for 10 s, 58 ˚C for 30 s and 72 ˚C for 20 s. The relative gene expression was analyzed calculated by the qPCRsoft3.2. The 18S ribosomal RNA gene was used as the internal reference. The primers of each gene were listed in S2 Table.

## Results and discussion

### Overview of the transcriptome sequence data

We obtained 39.4–63.6 million paired-end reads for 15 sample types in triplicates (45 libraries in total) (S3 Table). Of the quality-filtered reads 58.3–71.2% mapped to the reference genome of *F. filiformis* (S3 Table). The moderate mapability may be caused by the different strains used in this study compared with the reference genome from strain KACC42780 [20]. Although we expect this to not influence our results, it may cause lower sensitivity in faster evolving or strain specific genes. Differences in read mapability were not found between the M1 and XR strain which made the transcriptome comparable across these two strains (S3 Table). To validate the results of the RNA-seq analysis, 18 genes were randomly selected for quantitative real-time PCR (qRT-PCR). These genes showed expression patterns similar to those in the RNA-seq data (S1 Fig), indicating that our transcriptome sequencing provided a good estimate of gene expression patterns in the analyses of fruiting body development and heat stress response of *F. filiformis*.

### Temporal- and spatial-gene expression across *F. filiformis* development

Differential expression analysis indicated the highest number of differentially expressed genes (DEGs) in the transition from vegetative mycelium to primordium (1,313 up-regulated, 1,243 down-regulated), followed by young to mature fruiting body cap (1,284 up-regulated, 986 down-regulated) relative to primordium (Fig 1A; S2A Fig, S4 Table). This gene expression pattern was also recognized by the WGCNA analysis, a systems biology approach aimed at uncover gene modules which share gene expression patterns at a pre-specified similarity cut-off [47]. We identified six gene modules highly correlated with a single tissue type (Fig 2A; S5 Table). Among them, the primordium module (module no. 6), young fruiting body cap module (module no. 4) and mature fruiting body cap module (module no. 12) contained the highest number of genes (Fig 2A). These results indicate that primordium stage comprises the most significant morphogenetic transition, and that hymenium maturation and sporulation in young and mature fruiting body caps may also harbor complex molecular mechanisms. This gene expression pattern is consistent with those found in other Agaricomycetes [8, 22, 13, 49]. Moreover, the DEGs related to each developmental stage were enriched for GO terms typical for fruiting body formation, see S3 Fig and S6 Table.

### Primordium development includes genes widely conserved in Agaricomycetes

The top 20 up-regulated genes induced in primordium relative to vegetative mycelium were listed in Table 1. Building on a previously published dataset [13], we found these genes were widely conserved in fungi, and re-emphasized that several primordium-upregulated genes

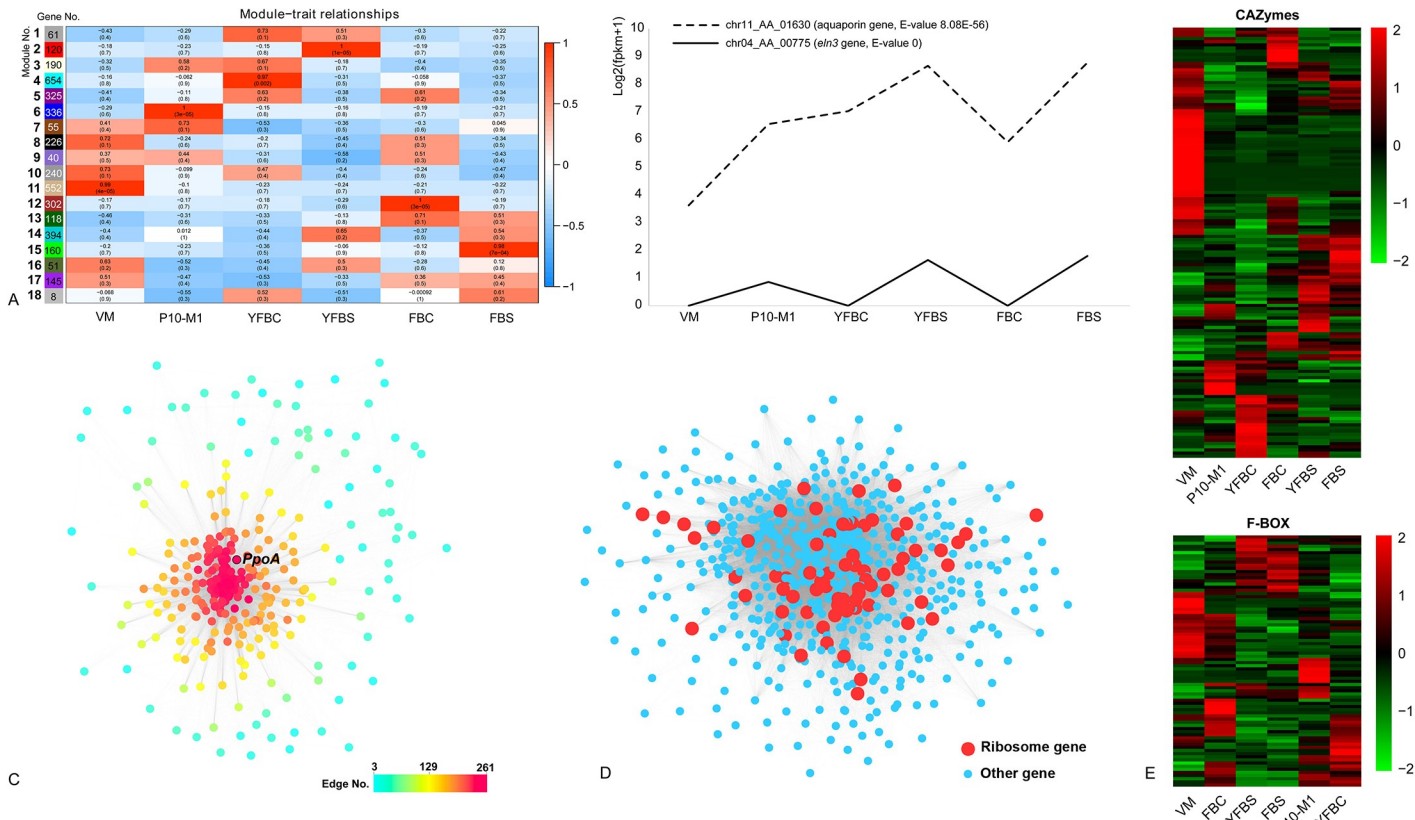

**Fig 2. Gene expression patterns in fruiting body development of *F. filiformis*.** A: Gene module-sample association revealed by gene co-expression analysis in WGCNA. Each row corresponds to a module, each column corresponds to a developmental stage or tissue type. VM, P10-M1, YFBC, YFBS, FBC and FBS correspond to vegetative mycelium, primordium, young fruiting body cap, young fruiting body stipe, fruiting body cap and fruiting body stipe, respectively. Upper and lower numbers in the cells indicate the correlation coefficient between the module and sample and the significance of the correlation (p-value), respectively; B: Expression patterns of the three aquaporin genes and the eln3 gene in each developmental stage of strain M1. E-value was reported by BLASTX search; C: Gene co-expression networks of the primordium module (module no. 6). The scale bar indicates the number of connections a gene has; D: Gene co-expression network of young fruiting body cap module (module no. 4). The red dots represent ribosomal protein encoding gene, blue dots represent other gene; E: Heatmap of the CAZyme and F-BOX gene expression in each developmental stage.

have homologs in simple multicellular or unicellular fungi. This indicates that some conserved gene families were recruited for complex multicellularity during evolution [13]. Among them, the Gti1/Pac2 family is conserved in all fungi (Table 1; Fig 3). Previously, this family has been discussed mostly in yeasts and pathogenic fungi, where it plays an important role in fungal growth and development [50]. Recent research revealed that this family is also developmentally regulated in *Armillaria ostoyae*, *Coprinopsis cinerea*, *Lentinus tigrinus*, *Rickenella mellea*, *Schizophyllum commune*, and *Phanerochaete chrysosporium* [13]. Thus, these genes may also play a key role in fruiting body development in Agaricomycetes. In addition, three TFs in the top 20 up-regulated genes (Zinc finger, C2H2 type, Zinc finger, Ring type, and Zn (2)-C6 fungal type) are conserved in Dikarya and Zoopagomycota plus later diverging phyla (Table 1; Fig 3), which reinforces the role of these TFs in complex multicellularity in fungi [13].

We found the gene encoding flammutoxin was conserved in 19 species in Agaricomycetes and 1 species in the Dacrymycetes (Table 1; Fig 3). This protein has been studied in *F. filiformis*, and may form a pore in the intestinal epithelial cells of fungivorous animals, leading to cell death [51]. However, because this protein is heat-labile, clinical reports about the intestinal dysfunction caused by ingestion of this mushroom are rare. Pore-forming proteins have been

**Table 1. Top 20 most upregulated and two other notable genes up-regulated in the primordium stage relative to vegetative mycelium.** Aquaporin and hydrophobin genes mentioned in the text are also shown.

| Rank | Protein ID | Log2 (FC) | P-value (FC) | Best Hit (Accession No.) | |
|---|---|---|---|---|---|
| 1 | chr11_AA_00208 | 10.37 | $5.8 \times 10^{-14}$ | - | orange |
| 2 | chr08_AA_01205 | 9.39 | $5 \times 10^{-55}$ | Hypothetical protein | blue |
| 3 | chr10_AA_00968 | 8.87 | $4.3 \times 10^{-60}$ | Short-chain dehydrogenase (IPR002347) | orange |
| 4 | chr11_AA_00046 | 8.70 | $1.9 \times 10^{-138}$ | Gti1/Pac2 family (IPR018608) | red |
| 5 | chr10_AA_00489 | 8.65 | $5 \times 10^{-12}$ | Flammutoxin (BAA32792) | green |
| 6 | chr01_AA_00267 | 8.62 | $7.4 \times 10^{-25}$ | - | yellow |
| 7 | chr03_AA_00235 | 8.52 | $7.5 \times 10^{-30}$ | - | yellow |
| 8 | chr08_AA_01207 | 8.37 | $7.9 \times 10^{-16}$ | - | cyan |
| 9 | chr08_AA_01206 | 8.35 | $1.3 \times 10^{-13}$ | - | orange |
| 10 | chr11_AA_00874 | 8.28 | $5.2 \times 10^{-18}$ | *Schizophyllum commune* hydrophobin, Sc3 (P16933) | orange |
| 11 | chr11_AA_01512 | 8.24 | $9.2 \times 10^{-7}$ | Cytochrome P450 (IPR001128) | red |
| 12 | chr07_AA_00932 | 8.22 | $6 \times 10^{-21}$ | Zinc finger, RING-type (IPR001841) | orange |
| 13 | chr03_AA_00276 | 7.97 | $8.1 \times 10^{-95}$ | *Flammulina velutipes* hydrophobin, fv-hyd1 (AB126686) | orange |
| 14 | chr04_AA_00509 | 7.91 | $5.4 \times 10^{-85}$ | Zinc finger, C2H2 (IPR007087) | yellow |
| 15 | chr05_AA_00568 | 7.77 | $6.2 \times 10^{-126}$ | Kre9/Knh1 family (IPR018466) | orange |
| 16 | chr08_AA_00570 | 7.01 | $1 \times 10^{-127}$ | - | orange |
| 17 | chr01_AA_00480 | 7.68 | $1.4 \times 10^{-17}$ | - | orange |
| 18 | chr08_AA_01181 | 7.56 | $1.3 \times 10^{-117}$ | - | orange |
| 19 | chr09_AA_01255 | 7.53 | $1.2 \times 10^{-26}$ | Zn(2)-C6 fungal-type (IPR001138) | orange |
| 20 | chr10_AA_01153 | 7.47 | $9 \times 10^{-17}$ | - | orange |
| 47 | chr11_AA_01264 | 5.81 | $3 \times 10^{-79}$ | Aquaporin (P43549) | orange |
| 54 | chr05_AA_00590 | 5.63 | $1 \times 10^{-9}$ | *Flammulina velutipes* hydrophobin, fv-hyd1 (AOV80987) | orange |

FC: Fold Change; Chytridio: Chytridiomycota; Mucoro: Mucoromycota; Zoopago: Zoopagomycota.

Genes conservation level in Fungi

Black-Species specific

Blue-Conserved in Physalacriaceae

Light blue-Conserved in Agaricomycetes

Green-Conserved in Basidiomycota

Yellow-Conserved in Dikarya

Orange-Conserved in Chytridio/Mucoro/Zoopago+higher

Red-Conserved in Fungi

studied in *Pleurotus* species, which exhibit cytotoxicity toward insect cells via pore formation in cell membranes to defend predation [52, 53]. Thus, whether flammutoxin serves to protect the fruiting body of *F. filiformis* predated by mammals or insects needs further investigation. In addition, a ricin-B lectin gene (chr11_AA_01461), showed high expression in fruiting bodies. It is homologous to the *Macrolepiota procera mpl*, Mpl protein shows toxicity towards the nematode *Caenorhabditis elegans* [54]. A re-analysis of its homologs in *C. cinerea*, *L. tigrinus*, *R. mellea*, *S. commune*, and *P. chrysosporium* revealed that it has fruiting body specific expression pattern in these species as well. This is in concordance with its expression pattern in *M. procera*, indicating a conserved role of this gene in protecting fruiting bodies against predators and parasites in Agaricomycetes [54]. Compared with the conserved flammutoxin and ricin-B lectin gene discussed here, we found the previously mentioned widely conserved ribotoxins gene is not exists in *F. filiformis* genome [55].

Interestingly, the second most up-regulated gene in primordium was restricted to the family Physalacriaceae, which indicated a role in primordium formation in this family (Table 1;

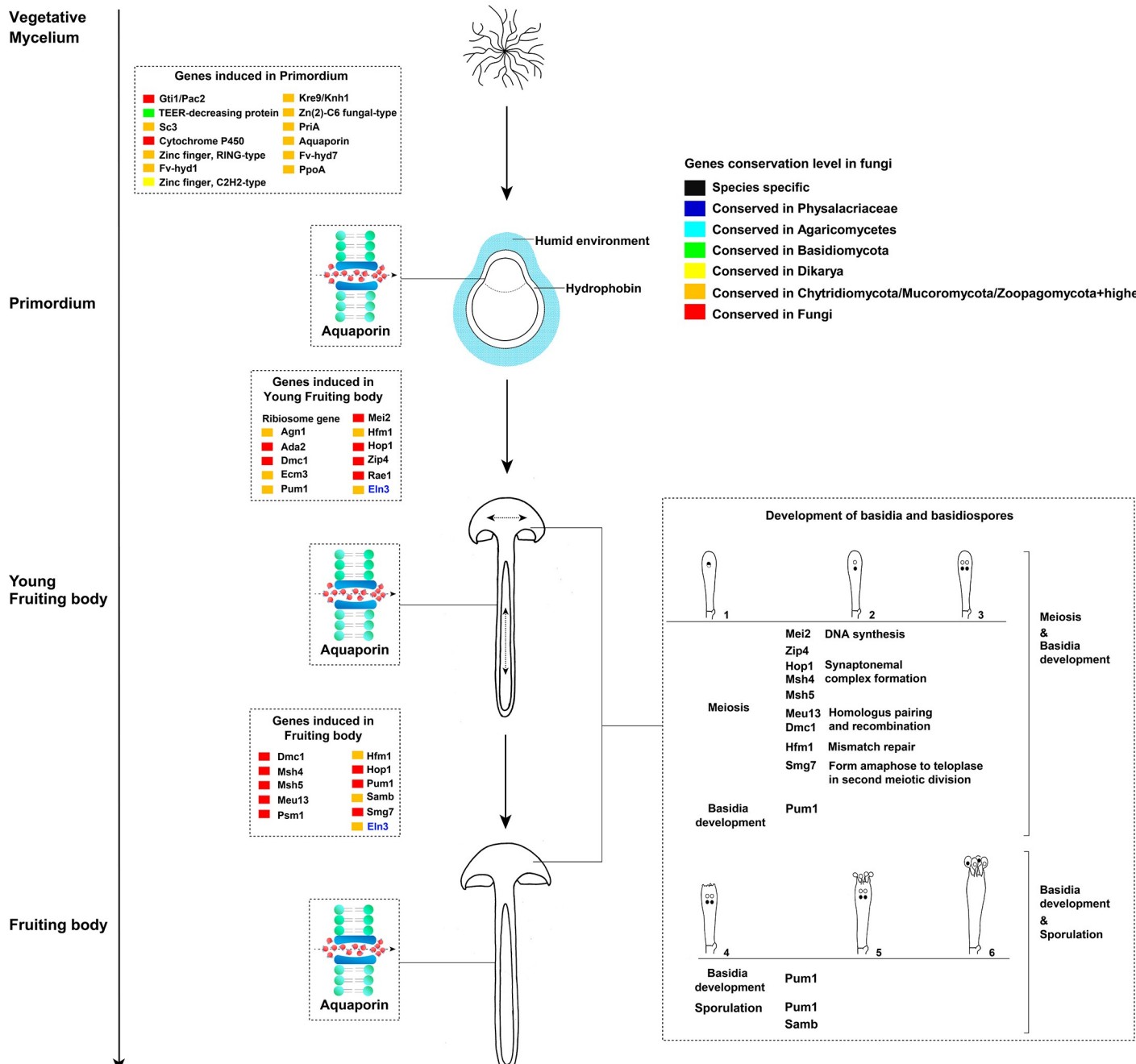

**Fig 3. Synoptic summary of key genes at different stages of development of *F. filiformis*.** 1–6 indicates developmental stages of basidia.

Fig 3). Unfortunately, our knowledge about this gene is limited. However, recent study reported the successful overexpression and RNA interference of the transcription factor *pdd1* in *F. filiformis*, which provides genetic tools to study this gene in the future [56]. Based on our analysis, almost half of the top 20 up-regulated genes in primordium stage without annotations (Table 1), they are widely conserved in simple multicellular and complex multicellular fungi, indicating that systematic studies are needed on mushroom development to help to understand multicellularity.

## Notable genes during fruiting body development

In the primordium stage, we found that genes encoding hydrophobins were (homologs of *fv-hyd1*, *fv-hyd7* and *S. commune sc3*) significantly up-regulated relative to vegetative mycelium (Fig 3; S2A Fig; S4 Table). It is well known that hydrophobins and cerato-platanins assemble at the hyphae surface to promote their aggregation in humid environments [57–59]. Hydrophobins may also hinder water absorption through the membrane [60].

We detected the aquaporin gene specifically induced in primordia and stipe tissues (Fig 2B; S2A Fig; S4 Table). Aquaporins are integral membrane proteins responsible for water and solute transport, and also involved in mycorrhizal formation and plant-fungal interactions during symbiosis establishment [26, 61]. Recent study revealed that aquaporins were also developmentally regulated in *L. bicolor* fruiting bodies [26]. However, aquaporins have been discussed mostly in the context of mycorrhizal fungi [62–65]. Based on our results, it seems likely that aquaporin-dependent water transport is a key process during mushroom development in saprotrophic fungi too, possibly in water transport along the stipe to facilitate water supply of the developing cap and gills [64]. A re-analysis of the data published by Sipos et al. (2017) [22] and Krizsán et al. (2019) [13] indicated that aquaporins were developmentally regulated in all six species studied by these authors, indicating that the role of aquaporins in development is not restricted to *Flammulina*, but may be widely conserved in fruiting body development. Aquaporins were differentially expressed in mycorrhizal species [2, 26, 64], which provides additional support to the hypothesis that fruiting bodies and ectomycorrhizae have many shared gene expression patterns, possibly pointing to common developmental origins [2, 64, 66].

Interestingly, we found that the *Flammulina* homolog of the stipe elongation gene *eln3* of *C. cinerea* possessed similar expression pattern to those of aquaporin genes during fruiting body development (Figs 2B and 3; S2A Fig; S4 Table). The mutant strain of this gene in *C. cinerea* produced aberrant fruiting bodies, in which the stipe hardly elongated during development [67]. The *eln3* homolog of *Volvariella volvacea* was also reported to be differentially expressed during fruiting body development [68]. A re-analysis indicated that *eln3* in *C. cinerea* and its homologs in *L. tigrinus*, *A. ostoyae* and *R. mellea* were developmentally regulated in RNA-Seq on data from previous studies [22, 13]. Homologs in *S. commune* and *P. chrysosporium* were not developmentally expressed, which might be explained by the lack of a stipe in these species. The broad conservation and expression patterns of *eln3* suggests that the molecular mechanisms of stipe elongation may be shared in Agaricomycetes, despite the independent origins of pileate-stipitate fruiting bodies in the class [28]. These results further highlight this gene as an interesting target in future strain improvement programs.

We found that one of the hub genes in the primordium module (module no. 6, WGCNA co-expression analysis) was a homolog of *Aspergillus nidulans ppoA* (Figs 2C and 3). This gene participates in oxylipin synthesis, which modulates sexual and asexual development in *A. nidulans* [69]. During sexual development, the PpoA protein initially localized in Hülle cells formed at the stage of cleistothecial primordium formation, and subsequently in immature cleistothecia in *A. nidulans* [69]. Over-expression of this gene in *A. nidulans* promotes sexual spore formation [69]. A re-analysis of this gene's homologs in *C. cinerea*, *R. mellea*, and *A. ostoyae* revealed a conserved expression pattern in these species, which indicates that oxylipins may mediate sexual development in the Ascomycota and the Agaricomycetes.

The homolog of *Lentinula edodes priA* was highly induced in primordium and young fruiting body cap of *F. filiformis* (S2A Fig; S4 Table). This gene was reported to possess the higher expression level in primordium and young fruiting body of *L. edodes* and over-expressing the *priA* gene in *L. edodes* monokaryotic mycelium remarkably decreased zinc ion accumulation, which indicates this gene may play a role in regulation of the intracellular zinc concentration

[70]. Surprisingly, we found its homologs were highly expressed in vegetative mycelium and lower expressed in fruiting bodies in *A. ostoyae*, *C. cinerea*, *L. tigrinus*, *R. mellea*, *S. commune*, and *P. chrysosporium*.

A large number (74) of ribosomal protein encoding genes, and homologs of genes involved in cell differentiation and cell wall formation in *S. pombe* (*agn1* and *rae1*), *Candida albicans* (*ada2*), and *S. cerevisiae* (*ecm3*) were hub genes in the young fruiting body cap module of the WGCNA analysis or were up-regulated relative to the primordium stage (Figs 2D and 3; S2A and S2B Fig; S4 Table). These results might reflect intense growth and protein synthesis in young fruiting bodies. Although the hymenium was immature in this stage, four meiosis regulation genes homologous to *S. pombe mei2*, *Cryptococcus neoformans dmc1*, and *S. cerevisiae hfm1*, *hop1*, and *zip4* were up-regulated (Fig 3; S2A Fig; S4 Table). Among them, *dmc1*, *hop1* and *hfm1* homologs were also up-regulated in mature cap (Fig 3; S2A Fig; S4 Table).

Compared to young fruiting body cap, more meiosis genes (homologs of *S. pombe psm1* gene, *C. neoformans dmc1* gene, and *S. cerevisiae*, *msh4*, *msh5*, *meu13*, *hop1*, *hfm1*, and *smg7*) were induced in the fruiting body cap (Fig 3; S2A Fig; S4 Table). Among the genes induced in young fruiting body cap and fruiting body cap, the genes homologous to *C. neoformans* pum1 may be noteworthy (Fig 3; S2A Fig; S4 Table). Pum1 is an RNA binding protein, and possesses an important role in post-transcriptional regulation in basidium development and sporulation in *C. neoformans* [71–73]. Previous studies revealed that the knockout of this gene in *C. neoformans* resulted in a severe defect in basidium formation [71–73]. In this study, we detected five genes homologous to *C. neoformans* pum1, two of them were developmentally regulated in young fruiting body cap and fruiting body cap, which indicates they may participate in basidium formation and sporulation in *F. filiformis* (Fig 3; S2A Fig). We detected another sporulation-related gene, homologous to *A. nidulans samB* (Fig 3; S2A Fig). Knock out of this gene in *A. nidulans* hindered ascospore formation [74]. A re-analysis of the homologs of pum1 and *Samb* in *A. ostoyae*, *C. cinerea*, *L. tigrinus*, *R. mellea*, *S. commune*, and *P. chrysosporium* revealed they possess conserved expression patterns in these species. Due to these two genes were widely conserved (Fig 3), we therefore speculate that some molecular mechanisms of spore formation may be conserved in fungi.

## CAZymes and F-box genes

Certain CAZymes were shown or assumed to participate in cell wall remodeling during fungal tissue differentiation [13, 49, 75–77]. We annotated 407 CAZymes genes in *F. filiformis*, 137 of them were differentially expressed (Fig 2E; S7 Table), which is consistent with previous studies in other mushroom-forming fungi [13, 20, 49]. Among these genes, Glycoside hydrolases (GH) and Glycosyltransferases (GT) were most abundant, with 57 and 30 genes, respectively. Although the targets of these families in fruiting bodies are currently unknown, the stage specific expression of these genes during fruiting body development reinforces the view that cell wall remodeling is a widespread and well-organized process in fruiting body development in Agaricomycetes.

F-box proteins play a key role in protein ubiquitination and modification, and are involved in many important biological processes not only in plants, but all eukaryotic [78, 79]. They were recently reported in relation to fruiting body development [13, 49]. In this study, 210 F-box encoding genes were annotated in *F. filiformis*, of which 80 were developmentally expressed and showed stage-specific expression patterns (Fig 2E; S8 Table). Similar expression patterns were also recognized in *A. ostoyae*, *C. cinerea*, *R. mellea*, *L. tigrinus*, and *S. commune*, which suggests that F-box genes may be crucial during fruiting body development in Agaricomycetes [13].

## Strain M1 expressed a large gene pool in response to heat stress relative to XR

Based on cultivation tests, we found that the growth of M1 strain showed no difference in 10˚C and 18˚C, while, the growth in XR strain was obviously retarded at 18 ˚C. On the molecular level, we found more differentially expressed genes in M1 (882 DEGs) than in XR (556 DEGs) (Fig 1C; S4 Fig). Based on the Venn diagram on Fig 1, we found two genes with elevated expression level in all developmental stages of strain M1 at 18 ˚C: a DNA damage repair gene homologous to *S. cerevisiae rad18* and a epoxide hydrolase gene homologous to *A. niger*. For strain XR, seven genes had an elevated expression level at all developmental stages under 18 ˚C (Fig 1C): two hsp20 genes, one WD repeat-containing gene, and four genes without annotation. Consistent with these functions, a GO enrichment analysis revealed that up-regulated genes in M1 were mainly enriched in 'response to stress' (GO:0006950, P<0.05), 'protein folding' (GO:0006457, P<0.01), and 'chaperone binding' (GO:0051087, P<0.05) etc. And the genes up-regulated in XR were mainly enriched in 'protein binding' (GO:0005515, P<0.01), 'DNA binding' (GO:0003677, P<0.01), and 'protein kinase binding' (GO:0019901, P<0.01) etc. (S5 Fig; S9 Table). These results indicate the different heat stress response strategies were employed in these two strains.

Specifically, differential expression analyses revealed that the M1 strain had more heat shock protein genes up-regulated than XR (S4 Fig). Among them, homologs of *S. cerevisiae hsp70*, *S. pombe hsp90* and another gene homologous to *Ustilago maydis fes1* induced in M1 at 18˚C may be noteworthy. The Hsp70 protein could protect nascent polypeptides and refold the damaged proteins under heat stress conditions [80]. If protein folding fails with Hsp70, Fes1 could interact with misfolded proteins and lead to their destruction by the ubiquitin-proteasome machinery [81]. Compared with Hsp70, Hsp90 functions primarily in the final maturation of proteins. Therefore, these genes may act as an "assembly line" [80] of protein maturation under heat stress during primordium development of strain M1. The heat stress induction of *hsp70* and *hsp90* was also reported in *Lentinula edodes* and *Ganoderma lucidum* [82, 83]. Homologs of these two genes were not differentially expressed in XR strain. Instead, the genes homologous to *S. pombe hsp20* were up-regulated in XR strain in all developmental stages at 18˚C, they play different roles than *hsp70* and *hsp90*, which are probably required to prevent misfolded protein aggregation and their degradation under heat stress [80].

In addition, the DNA damage repair gene homologous to *S. cerevisiae* rad18 was up-regulated in all developmental stages in M1. Yeast strains lacking Rad18 proteins may be highly sensitive to a wide variety of DNA damaging agents such as UVC-light, ROS stress and γ-radiation [84–86]. However, this gene was only up-regulated in the young fruiting body stage of XR grown at 18˚C. This example, combined with the fewer DEGs in the XR strain may indicate the loss of an ancestral heat stress response mechanisms in the commercial XR strain. This may result from the lack of temperature fluctuations and stress in general in factory settings.

## Conclusions

This study broadened our knowledge of fruiting body development and heat stress response of mushroom-forming fungi based on comparisons of transcriptomic data in *F. filiformis*. We detected a series of genes (e.g. aquaporins, *eln3*-homologs, hydrophobins, conserved transcription factors, oxylipin biosynthesis genes) that show conserved, dynamic expression during fruiting body development, and also uncovered signal for defense against high temperature in the heat tolerance strain (M1) (e.g. *hsp70*, *hsp90* and *fes1* homologs). These, or other differentially expressed genes, might be good candidates for in-depth experimental follow-up analyses

(e.g. gene knockout) to understand their specific roles and answer important or interesting questions that remained open. Analyzing the function of conserved genes in model and non-model species will be necessary to broaden our knowledge on fruiting body development in the Agaricomycetes.

## Supporting information

**S1 Fig. Correspondence between FPKM value and quantitative real-time PCR expression values for 18 randomly selected genes.** VM, P10-M1, P18-M1, YFBC, YFBS, FBC and FBS correspond to vegetative mycelium, primordium grown at 10°C, primordium grown at 18°C, young fruiting body cap, young fruiting body stipe, fruiting body cap and fruiting body stipe, of strain M1. P10-XR, P18-XR correspond to primordium grown at 10°C and 18°C of strain XR. Bar chart represents the FPKM values (left vertical axis), line chart represents the real-time PCR expression values (right vertical axis).
(JPG)

**S2 Fig. Gene expression pattern during fruiting body development.** A: Volcano plots of differential expression analysis for each comparison group; B: Gene co-expression network of the young fruiting body cap module (module no. 4 in Fig 2A). The scale bar indicates the number of connections a gene has.
(JPG)

**S3 Fig. GO enrichment of the genes up-regulated in each developmental stage.** X-axis indicates the ratio of the number of test genes and reference genes; Y-axis indicates the description of the functional terms.
(JPG)

**S4 Fig. Volcano plots of differential expression analysis of each comparison group of M1 and XR strain grown at 10°C and 18°C.**
(JPG)

**S5 Fig. GO enrichment of the genes up-regulated in each developmental stage grown at 18°C relative to 10°C in M1 and XR strain respectively.** X-axis indicates the ratio of the number of test genes and reference genes; Y-axis indicates the description of the functional terms.
(JPG)

**S1 Table. Gene expression pattern in each stage.**
(XLS)

**S2 Table. qRT-PCR primers used in this study.**
(XLS)

**S3 Table. Total reads and mapping rates of each sample.**
(XLS)

**S4 Table. DEGs in each developmental stage.**
(XLS)

**S5 Table. Genes involved in each gene module based on WGCNA analysis.**
(XLS)

**S6 Table. Go enrichment of the DEGs in each successive developmental stage.**
(XLS)

**S7 Table. The CAZymes genes annotated in *F. filiformis*.**
(XLS)

**S8 Table. F-Box genes annotated in *F. filiformis*.**
(XLS)

**S9 Table. GO enrichment of the DEGs in each developmental stage of M1 and XR grown at 18°C relative to 10°C respectively.**
(XLS)

## Acknowledgments

The authors are grateful to Prof. Won-Sik Kong (Mushroom Research Division, National Institute of Horticultural and Herbal Science, Rural Development Administration, Eumsung, Republic of Korea) for generously sharing the genome sequencing data of the strain KACC42780 for us. Ms. Ming-Li Li (Kunming Institute of Zoology, Chinese Academy of Sciences), Dr. Botond Hegedüs and Dr. Balázs Bálint (Synthetic and Systems Biology Unit, Institute of Biochemistry, Biological Research Centre, Szeged), Dr. Ti-Cao Zhang (Yunnan University of Chinese Medicine), Dr. Peter Langfelder (University of California), Dr. Yong-Ping Fu (Jilin Agricultural University) and Dr. Robin Ohm (Utrecht University) were acknowledged for their kind help in data analyses. The anonymous reviewers are also gratefully acknowledged for their comments and suggestions.

## Author Contributions

**Conceptualization:** Xiao-Bin Liu, En-Hua Xia, Jian-Ping Xu, László G. Nagy, Zhu L. Yang.

**Formal analysis:** Xiao-Bin Liu, En-Hua Xia, Meng Li, Pan-Meng Wang, László G. Nagy.

**Investigation:** Xiao-Bin Liu, Yang-Yang Cui, Jing Li.

**Resources:** Jin-Xia Zhang, Bao-Gui Xie, Jun-Jie Yan.

**Supervision:** Zhu L. Yang.

**Writing – original draft:** Xiao-Bin Liu.

**Writing – review & editing:** László G. Nagy, Zhu L. Yang.

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
