## [Decision Letter · Decision Letter 0]

10 Apr 2020

PONE-D-20-02460

Transcriptome data reveal conserved patterns of fruiting body development and response to heat stress in the mushroom-forming fungus Flammulina filiformis

PLOS ONE

Dear Dr. Liu,

Thank you for submitting your manuscript to PLOS ONE. After careful consideration, we feel that it has merit but does not fully meet PLOS ONE’s publication criteria as it currently stands. Therefore, we invite you to submit a revised version of the manuscript that addresses the points raised during the review process.

The three expert reviewers found much that they liked about your study.  However, all three had several suggestions for improvement. Reviewer 1 had several valuable comments about data interpretation and presentation. Reviewer 2 has well-founded concerns about the availability of the fungal strains used to produce the data in the paper. These need to be addressed prior to publication. Reviewer 3 makes several compelling arguments for a rebalancing of the paper to better introduce the project and then interpret the findings in a broader context.  All three reviewers make many other suggestions that should be considered during the revision of your manuscript.

We would appreciate receiving your revised manuscript by June 1, 2020. To enhance the reproducibility of your results, we recommend that if applicable you deposit your laboratory protocols in protocols.io, where a protocol can be assigned its own identifier (DOI) such that it can be cited independently in the future. For instructions see: http://journals.plos.org/plosone/s/submission-guidelines#loc-laboratory-protocols

We look forward to receiving your revised manuscript.

Kind regards,

Katherine A. Borkovich, Ph.D.

Academic Editor

PLOS ONE

Journal Requirements:

Reviewers' comments:

Reviewer's Responses to Questions

**Comments to the Author**

1. Is the manuscript technically sound, and do the data support the conclusions?

Reviewer #1: Yes

Reviewer #2: Yes

Reviewer #3: Yes

2. Has the statistical analysis been performed appropriately and rigorously? 

Reviewer #1: Yes

Reviewer #2: N/A

Reviewer #3: Yes

3. Have the authors made all data underlying the findings in their manuscript fully available?

Reviewer #1: Yes

Reviewer #2: No

Reviewer #3: Yes

4. Is the manuscript presented in an intelligible fashion and written in standard English?

Reviewer #1: Yes

Reviewer #2: Yes

Reviewer #3: Yes

5. Review Comments to the Author

Reviewer #1: The authors have performed a genome-wide gene expression analysis (using RNA-Seq) on several stages of mushroom development in the commercially interesting mushroom Flammulina filiformis. Furthermore, they compared gene expression in two strains with varying abilities of heat resistance. The resulting expression profiles were compared to previously published expression analyses, resulting in candidate genes involved in development and heat resistance. No follow-up functional analyses were performed to study these genes. However, since this is not a traditional genetic model system, this would have been challenging to do. The expression profiles generated by this study will be a useful addition to the set of comparative mushroom-development gene expression data sets. The analyses appear to have been performed correctly, and the conclusions are supported by the data.

Line 198. Read mapping percentages are very low (58.3-71.2%), presumably because the reference assembly is from a different strain. Although this may be hard to avoid in this study, it is still important to realize possible implications. For example, the expression profiles of faster evolving or strain-specific genes will be captured less accurately, since sequence similarity will be less. Please consider adding a few sentences describing this.

Several Excel files/sheets are provided with the developmentally expressed genes. It would be beneficial to the readers to also include a sheet with the expression of all the genes, whether they are differentially expressed or not. Moreover, as far as I can see no such files are provided for the expression values during the heat/cold experiment (maybe I overlooked it). Please provide the full expression set for those conditions as well.

Line 250. Three transcription factors are mentioned. Do these include any of the known regulators of mushroom development that were previously identified in Schizophyllum commune (e.g. c2h2, fst4, etc).

Line 211. Please consider introducing the concept of WGCNA here in a few sentences, explaining the goal to the readers of PLOS.

Line 249. “this TF”. It seems that this refers to “Gti1/Pac2 family (IPR018608)” mentioned in Line 244 and Table 1. Is this indeed a transcription factor (i.e. DNA-binding) or is it more generally a ‘regulator’?

Line 99. “Previous studies, revealed protein expression patterns”. The comma is not needed here. Moreover, more accurate would be: “gene expression patterns”

Line 166. The name of the tool is Hisat.

Reviewer #2: Reviewer’s comments for the author

This manuscript of Liu et al. describes a transcriptomic analysis on the agaric Flammulina filiformis (formely F. velutipes). The manuscript is generally written in concise well understandable English.

Specific comments:

- Abstract: The authors speak of >200 reference genomes which were analysed to fish for conserved genes playing a role in mushroom formation. However, they seem not to explain this in detail further down in the manuscript, e.g. in the Material and methods section.

- Introduction:

o Attention is paid to the two traditional agaric model systems Schizophyllum commune and Coprinopsis cinerea in which advances in molecular genetics techniques allowed to study of fruiting body formation relevant genes. In addition, the authors speak of economically important non-model systems in scientific focus since more recently but do not cite the worldwide cultivated edible mushroom Cyclocybe aegerita (poplar mushroom; formerly Agrocybe aegerita) which in this context, besides its economic importance, seems not only interesting enough to mention for its capability of haploid fruiting without mating and bioactive compound production but, since recently, also allows functional genetics analyses (MGG 294:663-677. https://doi.org/10.1007/s00438-018-01528-6; AEM 85:e01549-19. https://doi.org/10.1128/AEM.01549-19; Beilstein J. Org. Chem. 2019, 15, 1000–1007. doi:10.3762/bjoc.15.98).

o Apart from this shortcoming, the introduction is well-written.

- Material & Methods:

o The two investigated strains of F. filiformis do not seem to be available from an internationally accessible strain collection like the Westerdijk Institute (Utrecht, Netherlands) or the ATCC (American Type Culture Collection in Manassas, Virginia, United States). How will the authors guarantee that scientists outside China will not be refused official requests for F. filiformis M1 (CCMSSC04554) and F. filiformis XR for research purposes (independent from collaboration with Chinese scientists) in view of the very restrictive Chinese policy on national bioresources normally resulting in refusal of biomaterial issuance to scientists outside China?

o Furthermore, the reviewer thinks that details should be given on the exact geographic and, if available, the habitat origin of both strains (something like “…strain XR was isolated from a mushroom market…”, line 126, is hardly informative).

o Apart from that the reviewer suggests to correct every mentioning of fungal strains in this manuscript so far to always mention them together with the species name, e.g. F. filiformis M1 and F. filiformis XR.

o The reviewer sees one more shortcoming in this section (RNA-sampling and –sequencing was carried out accurately): the authors should specifically mention how they performed the realtime quantitative PCR (qRT-PCR) assessment of the differential expression of the 22 randomly selected fruiting- and heat-stress correlated genes they mention further down (Fig. S1). This is to include in a small paragraph on qRT-PCR in this section before the study can be accepted for publication in PLoS One.

In addition, the figure caption of Fig. S1 should be more informative (the authors should double-check on this also in their other figures): what will “chr11_AA_01521” or “P10-M1” tell the uninformed reader of the study? The reviewer is aware that fruiting stage describing abbreviations like “P10-M1” are spelled out further down in the figure legend of Fig. 2 but explaining these terms in in Fig. S1, too seems adequate to increase readability of the manuscript. Apart from that, there seems to be a discrepancy between the 22 genes checked with qRT-PCR in the manuscript text (line 201) and in the figure legend where it states 18 genes were qRT-PCR-checked (line 695)!

Terms like “chr11_...” and chr05…” also imply that the reads were potentially mapped to chromosome level of the F. filiformis genome. Could this please be specified somewhere in this section or elsewhere appropriate in the manuscript?

o Apart from the 3 shortcomings, this section seems fine to the reviewer.

- Results and Discussion:

o Line 268 and Tab. 1: It is mentioned that the flammutoxin-encoding gene is conserved (if this is a typo there "...conversed...", line 268; it should also be "...was..." instead of "...were..." I assume) in 20 Basidiomycota species. It seems interesting to compare whether other anti-antagonist mushroom compounds like galectins, ribotoxins, or other antimicrobial proteins/peptides are also conserved and expressed (When? In vegetative mycelium or during fruiting?) in F. filiformis? The authors should check on this as well and possibly complement their results with this recommendable checkup.

o Lines 278-280: Maybe the authors should like to shortly emphasise the need for a resilient functional genetics toolbox in F. filiformis that would greatly serve increasing knowledge about this interesting gene? In this context, they may like to discuss which tools (transformation techniques, transformant selection markers, overexpression tools,...) are available for F. filiformis.

o One general comment on gene notation: Either the authors follow the widely-applied “basidiomycete code” (wild type gene names written like in yeast code but all letters lowercase; protein names as in yeast code…) or the “yeast code” (wild type gene names written italicised in uppercase letters with a number; proteins non-italicised, first letter uppercase; this code is sometimes applied in Basidiomycota, too, especially when addressing gene names of a certain species like AaeAGT1 for the ribotoxin gene of the poplar mushroom to not get somewhat “inartfully” looking gene names like Aaeagt1). This should be corrected before publication.

Reviewer #3: This study examined mushroom forming fungi, with an emphasis on how the winter mushroom, Flammulina filiformis, development responds to heat stress. The authors used transcriptomics and compared >200 genomes to identify conserved genes that may play important roles in fruiting body development and heat stress. The study compared responses in heat sensitive and heat resistant strains of F. filiformis, and showed that the different strains responded differently to heat stress, as heat stress induced more genes in the heat resistant strain than in the heat sensitive strain. In their analyses, the authors highlighted a suite of conserved genes that were associated with primordium development, as well as revealed several novel fruiting specific genes.

I enjoyed this study and found their approach to thoughtful. However, I believe that this study, as written, may not appeal to the broad readership of PLOS One. Furthermore, I found their lines of evidence, and their inclusion of supporting literature to be vague. In the first few paragraphs of their Introduction, the authors did not effectively place their study within a broader context. Yet, their Results and Discussion did interpret the implications of previous studies, and how their work contributes to addressing gaps in our knowledge base, as well as contributes valuable data to the genetic underpinnings of fungal fruit body development.

I recommend that the authors re-work both their Introduction and Discussion (Results and Discussion) to place their study within a broader context. Please interpret the cited literature more extensively, and be more specific about the gaps in our knowledge about mushroom development in Agaricomycetes. What triggers fruit body / primordial development? How are these triggers found to elicit reproduction in natural systems? To what extent do mushroom cultivators emulate these processes within the built environment? Why does cultivation require low temperatures of this particular mushroom-- is it for coloration or cluster formation? What is the significance of trehalose biosynthesis genes, and what can it tell us about the mushrooms growing at different temperatures? Why might be expect aquaporins or hydrophobins to arise in these different growing conditions? In general, there needs to be greater rationale for the study and heightened detail about the genes, especially greater background on hypothesized mechanisms. In fact, I would have appreciated more explicitly stated hypotheses, with associated rationale, within the Introduction.

Even before going into great detail with their focal study organism, I recommend that the authors begin with a more general introduction about how organisms (across different trophic levels) respond to environmental stress. Additionally, I recommend that the authors emphasize what gene regulation, as inducible gene expression, could tell us about how bacteria and multicellular organisms, in general, promote their survival in stressful conditions, as well as faced with heat-stress, more specifically. Another approach that I recommend would be to bring in conceptual frameworks, to highlight your intervention within the broader fields, such as an ecological framework, (e.g., Grime’s pyramid, with an emphasis on stress-tolerant strategies), or framing this within an evolutionary context, in terms of selective pressures and eco-physiological and metabolic responses to such pressures. Then, once the rationale is more compelling, I would like the authors to pinpoint their focus on mushrooms, and then F. filiformis.

I realize that mushroom are important in the ecosystem, as the authors mention at the onset of their manuscript. I certainly appreciated the authors' discussion of Coprinopsis cinerea and Schizophyllum commune, as well as mentioning “ecologically or economically important non-model species.” Yet, why are: Agaricus bisporus, Flammulina filiformis, Armillaria ostoyae, Lentinula edodes and Lentinus tigrinus ecologically important? A little more discussion of this would add some much needed depth to introduce this study.

There were minor grammatical errors what interfered with the ease of readability of the manuscript. I recommend that the authors proofread the paper for grammar and flow. Define or limit terms introduced that may be industry specific. Given the broad audience at PLOS One, some terms may require additional clarifying details. For instance (line 135): is "scratched" an industry specific term to mimic disturbance -- leading to an increased investment in reproduction/fruiting to pass on genetic material to progeny?

Methodologically, this paper had many strengths. However, since the study was not hypothesis driven, it was challenging to determine how the research frames an intervention to fit into a broader context. I recommend that the authors look to cite studies using transcriptomics to examine fungal responses to experimental warming in the field, such as this study on meta-transcriptomics by Romero-Olivares et al. 2019 in Frontiers in Microbiology doi: 10.3389/fmicb.2019.01914, which does an excellent job of framing the stress response trade-offs with decay gene expression. I would recommend a greater discussion of these trade-offs within the sun-section in the Results/Discussion on the CAZymes and F-box genes (L 369-386).

Although I appreciated the discussion on aquaporins and hydrophobins (line 288-296), I would have liked greater linkages of to function, and specifically a greater discussion comparing and contrasting aquaporins expression within ectomycorrhizal fungi, beyond putative common developmental origins (line 304). Furthermore, I would recommend for these notable genes to be mentioned in the introduction, as well. In the introduction, trehalose synthase genes were mentioned, but then was not detected or discussed in the Discussion. Was this surprising that there were no trehalose synthase genes detected? Please adjust caption for Table 1 (Top 20 and two notable genes…), as that text is unclear. I would have like greater integration of the figures into the manuscipt. Include a list of figures, as the list of supplemental figures are given. When referring to the figure, it is challenging to find the information, required for interpretation. For instance, I recommend that you include legend in fig 1 that describes or identify the color schemes or components of the figures.

The authors adeptly discussed the results of the differential gene analyses, specifically in relation to heat-shock proteins and assembly line for protein maturation under heat stress. I found their results interesting and could provide an important basis for understanding developmental responses to heat stress. If the authors undertake a major revision of their Introduction and Discussion (Results and Discussion) sections, to increase broad-scale appeal, that I think this manuscript would be suitable for publication in PLOS One.

6. PLOS authors have the option to publish the peer review history of their article (what does this mean?). If published, this will include your full peer review and any attached files.

Reviewer #1: No

Reviewer #2: No

Reviewer #3: No

---

## [Author Response · Author response to Decision Letter 0]

28 Apr 2020

Dear Editors of PLoS One,

Thank you very much for your email with regard to our manuscript entitled “Transcriptome data reveal conserved patterns of fruiting body development and response to heat stress in the mushroom-forming fungus Flammulina filiformis” (PONE-D-20-02460). We found that the critiques, comments and suggestions of the reviewers are very constructive and helpful. We have revised the manuscript according to the comments and suggestions. We really appreciate your kind help. We believe that the revised manuscript is much improved and hope it will be acceptable for publication in PLoS One.

We look forward to your further communications with regard to this manuscript. Below are our responses to the comments raises by the reviewers.

Best wishes,

Xiao-Bin Liu, László G. Nagy, Zhu L. Yang

Reviewer #1: 

1) Line 198. Read mapping percentages are very low (58.3-71.2%), presumably because the reference assembly is from a different strain. Although this may be hard to avoid in this study, it is still important to realize possible implications. For example, the expression profiles of faster evolving or strain-specific genes will be captured less accurately, since sequence similarity will be less. Please consider adding a few sentences describing this.

Reply: Thank you very much for the suggestion, we add short comments in Lin224-229.

Several Excel files/sheets are provided with the developmentally expressed genes. It would be beneficial to the readers to also include a sheet with the expression of all the genes, whether they are differentially expressed or not. Moreover, as far as I can see no such files are provided for the expression values during the heat/cold experiment (maybe I overlooked it). Please provide the full expression set for those conditions as well.

Reply: Done.

Line 250. Three transcription factors are mentioned. Do these include any of the known regulators of mushroom development that were previously identified in Schizophyllum commune (e.g. c2h2, fst4, etc).

Reply: No, they are not orthologs of the genes identified in previously study in S. commune.

Line 211. Please consider introducing the concept of WGCNA here in a few sentences, explaining the goal to the readers of PLOS.

Reply: Done, we add the concept in line 241-242.

Line 249. “this TF”. It seems that this refers to “Gti1/Pac2 family (IPR018608)” mentioned in Line 244 and Table 1. Is this indeed a transcription factor (i.e. DNA-binding) or is it more generally a ‘regulator’?

Reply: It is more generally a regulator. In Schizosaccharomyces pombe, the Gti1 protein related to gluconate uptake, the Pac2 related to sexual development. We rephrased the text accordingly.

Line 99. “Previous studies, revealed protein expression patterns”. The comma is not needed here. Moreover, more accurate would be: “gene expression patterns”

Reply: Done, thank you very much.

Line 166. The name of the tool is Hisat.

Reply: Done, thank you very much.

Reviewer #2: 

- Abstract: The authors speak of >200 reference genomes which were analysed to fish for conserved genes playing a role in mushroom formation. However, they seem not to explain this in detail further down in the manuscript, e.g. in the Material and methods section.

Reply: Thanks for your suggestion. We add it in Material and methods section in Line 203-209.

- Introduction:

o Attention is paid to the two traditional agaric model systems Schizophyllum commune and Coprinopsis cinerea in which advances in molecular genetics techniques allowed to study of fruiting body formation relevant genes. In addition, the authors speak of economically important non-model systems in scientific focus since more recently but do not cite the worldwide cultivated edible mushroom Cyclocybe aegerita (poplar mushroom; formerly Agrocybe aegerita) which in this context, besides its economic importance, seems not only interesting enough to mention for its capability of haploid fruiting without mating and bioactive compound production but, since recently, also allows functional genetics analyses (MGG 294:663-677. https://doi.org/10.1007/s00438-018-01528-6; AEM 85:e01549-19. https://doi.org/10.1128/AEM.01549-19; Beilstein J. Org. Chem. 2019, 15, 1000–1007. doi:10.3762/bjoc.15.98).

Reply: Thank you very much. We cite this work in introduction section. Line88-91.

- Material & Methods:

o The two investigated strains of F. filiformis do not seem to be available from an internationally accessible strain collection like the Westerdijk Institute (Utrecht, Netherlands) or the ATCC (American Type Culture Collection in Manassas, Virginia, United States). How will the authors guarantee that scientists outside China will not be refused official requests for F. filiformis M1 (CCMSSC04554) and F. filiformis XR for research purposes (independent from collaboration with Chinese scientists) in view of the very restrictive Chinese policy on national bioresources normally resulting in refusal of biomaterial issuance to scientists outside China?

Reply: We are very willing to deposit our strain in relevant collection center. Indeed, one of the strains (F. filiformis M1, CCMSSC04554) in our study already deposited in China Center for Mushroom Spawn Standards and Control. We would like to deposit them in other international collection center after the pandemic of COVID-19.

o Furthermore, the reviewer thinks that details should be given on the exact geographic and, if available, the habitat origin of both strains (something like “…strain XR was isolated from a mushroom market…”, line 126, is hardly informative).

Reply: Done. Line136-139.

o The reviewer sees one more shortcoming in this section (RNA-sampling and –sequencing was carried out accurately): the authors should specifically mention how they performed the realtime quantitative PCR (qRT-PCR) assessment of the differential expression of the 22 randomly selected fruiting- and heat-stress correlated genes they mention further down (Fig. S1). This is to include in a small paragraph on qRT-PCR in this section before the study can be accepted for publication in PLoS One.

Reply: Done, we add it in materials and methods section, in Line210-Lin219.

In addition, the figure caption of Fig. S1 should be more informative (the authors should double-check on this also in their other figures): what will “chr11_AA_01521” or “P10-M1” tell the uninformed reader of the study? The reviewer is aware that fruiting stage describing abbreviations like “P10-M1” are spelled out further down in the figure legend of Fig. 2 but explaining these terms in in Fig. S1, too seems adequate to increase readability of the manuscript. Apart from that, there seems to be a discrepancy between the 22 genes checked with qRT-PCR in the manuscript text (line 201) and in the figure legend where it states 18 genes were qRT-PCR-checked (line 695)!

Reply: We add the legend in Fig. S1, the number of the genes involved in qRT-PCR should be 18. Thank you very much.

Terms like “chr11_...” and chr05…” also imply that the reads were potentially mapped to chromosome level of the F. filiformis genome. Could this please be specified somewhere in this section or elsewhere appropriate in the manuscript?

Reply: Yes, the reference genome was assembled in chromosome level, we mentioned it in Line176.

-Results and Discussion:

o Line 268 and Tab. 1: It is mentioned that the flammutoxin-encoding gene is conserved (if this is a typo there "...conversed...", line 268; it should also be "...was..." instead of "...were..." I assume) in 20 Basidiomycota species. 

Reply: Done. Thank you very much.

It seems interesting to compare whether other anti-antagonist mushroom compounds like galectins, ribotoxins, or other antimicrobial proteins/peptides are also conserved and expressed (When? In vegetative mycelium or during fruiting?) in F. filiformis? The authors should check on this as well and possibly complement their results with this recommendable checkup.

Reply: Thank you very much for this suggestion. We didn’t find the galectins and ribotoxins coding gene in F. filiformis genome. Whereas, we find Macrolepiota procera mpl homologous gene in F. filiformis. This gene is conserved in Agaricomycetes, and have a conserved expression pattern in fruiting body stage. We discussed its role in line285-lin292.

o Lines 278-280: Maybe the authors should like to shortly emphasise the need for a resilient functional genetics toolbox in F. filiformis that would greatly serve increasing knowledge about this interesting gene? In this context, they may like to discuss which tools (transformation techniques, transformant selection markers, overexpression tools,...) are available for F. filiformis.

Reply: Done, Line296-Line298. Thank you very much.

o One general comment on gene notation: Either the authors follow the widely-applied “basidiomycete code” (wild type gene names written like in yeast code but all letters lowercase; protein names as in yeast code…) or the “yeast code” (wild type gene names written italicised in uppercase letters with a number; proteins non-italicised, first letter uppercase; this code is sometimes applied in Basidiomycota, too, especially when addressing gene names of a certain species like AaeAGT1 for the ribotoxin gene of the poplar mushroom to not get somewhat “inartfully” looking gene names like Aaeagt1). This should be corrected before publication.

Reply: Done. Thank you very much.

Reviewer #3: 

I enjoyed this study and found their approach to thoughtful. However, I believe that this study, as written, may not appeal to the broad readership of PLOS One. Furthermore, I found their lines of evidence, and their inclusion of supporting literature to be vague. In the first few paragraphs of their Introduction, the authors did not effectively place their study within a broader context. Yet, their Results and Discussion did interpret the implications of previous studies, and how their work contributes to addressing gaps in our knowledge base, as well as contributes valuable data to the genetic underpinnings of fungal fruit body development.

Reply: Thanks for this suggestion. We think we already put our research within a broader context to discuss the complex multicellular evolution from fruiting body development. We also highlighted the importance of our work as uncover the fruiting body development mechanism which can help us to understanding the complex multicellular evolution. 

I recommend that the authors re-work both their Introduction and Discussion (Results and Discussion) to place their study within a broader context. Please interpret the cited literature more extensively, and be more specific about the gaps in our knowledge about mushroom development in Agaricomycetes. What triggers fruit body / primordial development? How are these triggers found to elicit reproduction in natural systems? To what extent do mushroom cultivators emulate these processes within the built environment? Why does cultivation require low temperatures of this particular mushroom-- is it for coloration or cluster formation? What is the significance of trehalose biosynthesis genes, and what can it tell us about the mushrooms growing at different temperatures? Why might be expect aquaporins or hydrophobins to arise in these different growing conditions? In general, there needs to be greater rationale for the study and heightened detail about the genes, especially greater background on hypothesized mechanisms. In fact, I would have appreciated more explicitly stated hypotheses, with associated rationale, within the Introduction.

Reply: Thanks for the suggestion, In order to be more specific about the gaps in our knowledge about mushroom development in Agaricomycetes, we interpret the previous work more extensively in introduction part. However, we intend to keep the introduction concise, therefore, we do not go into much detail on some the background of fruiting body induction, rather, we refer to recent reviews on the topic. Line76-line81, line83-line87, line93-line94, line111-line118.

Even before going into great detail with their focal study organism, I recommend that the authors begin with a more general introduction about how organisms (across different trophic levels) respond to environmental stress. Additionally, I recommend that the authors emphasize what gene regulation, as inducible gene expression, could tell us about how bacteria and multicellular organisms, in general, promote their survival in stressful conditions, as well as faced with heat-stress, more specifically. Another approach that I recommend would be to bring in conceptual frameworks, to highlight your intervention within the broader fields, such as an ecological framework, (e.g., Grime’s pyramid, with an emphasis on stress-tolerant strategies), or framing this within an evolutionary context, in terms of selective pressures and eco-physiological and metabolic responses to such pressures. Then, once the rationale is more compelling, I would like the authors to pinpoint their focus on mushrooms, and then F. filiformis.

Reply: Thanks for this suggestion, In order to keep compact and concise of the introduction, we included some general considerations on heat stress response in line117-line118.

I realize that mushroom are important in the ecosystem, as the authors mention at the onset of their manuscript. I certainly appreciated the authors' discussion of Coprinopsis cinerea and Schizophyllum commune, as well as mentioning “ecologically or economically important non-model species.” Yet, why are: Agaricus bisporus, Flammulina filiformis, Armillaria ostoyae, Lentinula edodes and Lentinus tigrinus ecologically important? A little more discussion of this would add some much needed depth to introduce this study.

Reply: Thanks for the suggestion, we revised this in line88-91, with the statement of the saprotrophic fungi, plant pathogen and ectomycorrhizal fungi.

There were minor grammatical errors what interfered with the ease of readability of the manuscript. I recommend that the authors proofread the paper for grammar and flow. Define or limit terms introduced that may be industry specific. Given the broad audience at PLOS One, some terms may require additional clarifying details. For instance (line 135): is "scratched" an industry specific term to mimic disturbance -- leading to an increased investment in reproduction/fruiting to pass on genetic material to progeny?

Reply: The “scratched” is used to describe the physical perturbation of the mycelium surface to stimulate fruiting body development.

Methodologically, this paper had many strengths. However, since the study was not hypothesis driven, it was challenging to determine how the research frames an intervention to fit into a broader context. I recommend that the authors look to cite studies using transcriptomics to examine fungal responses to experimental warming in the field, such as this study on meta-transcriptomics by Romero-Olivares et al. 2019 in Frontiers in Microbiology doi: 10.3389/fmicb.2019.01914, which does an excellent job of framing the stress response trade-offs with decay gene expression. I would recommend a greater discussion of these trade-offs within the sun-section in the Results/Discussion on the CAZymes and F-box genes (L 369-386).

Reply: Thank you very much for the recommended paper. This is a nice paper which investigated the tradeoff in fungal resource allocation under experimental warming treatments. However, in the CAZymes and F-box genes section, we mainly discussed the expression pattern of the CAZyme genes during fruiting body development, since this gene family recently reported not only involved in wood decay, but also related to fruiting body development. Therefore, we think the topic of the paper which reviewer recommends is different from our study, and we do not discuss it in our study.

Although I appreciated the discussion on aquaporins and hydrophobins (line 288-296), I would have liked greater linkages of to function, and specifically a greater discussion comparing and contrasting aquaporins expression within ectomycorrhizal fungi, beyond putative common developmental origins (line 304). Furthermore, I would recommend for these notable genes to be mentioned in the introduction, as well. In the introduction, trehalose synthase genes were mentioned, but then was not detected or discussed in the Discussion. Was this surprising that there were no trehalose synthase genes detected? Please adjust caption for Table 1 (Top 20 and two notable genes…), as that text is unclear. I would have like greater integration of the figures into the manuscipt. Include a list of figures, as the list of supplemental figures are given. When referring to the figure, it is challenging to find the information, required for interpretation. For instance, I recommend that you include legend in fig 1 that describes or identify the color schemes or components of the figures.

Reply: Thank you very much for the comments. We add functional discussion of the aquaporins in Line311-313. In our study, we didn’t detect the differential expression of the trehalose synthase genes. This may be interpreted as the heat treatment in our study was not severe enough, since the trehalose synthase expression often happens under severe heat stress, but may also reflect Flammulina-specific traits. We currently don’t have enough data to formulate sounds hypotheses on that.

---

## [Decision Letter · Decision Letter 1]

15 Jun 2020

PONE-D-20-02460R1

Transcriptome data reveal conserved patterns of fruiting body development and response to heat stress in the mushroom-forming fungus Flammulina filiformis

PLOS ONE

Dear Dr. Liu,

Thank you for submitting your manuscript to PLOS ONE. After careful consideration, we feel that it has merit but does not fully meet PLOS ONE’s publication criteria as it currently stands. Therefore, we invite you to submit a revised version of the manuscript that addresses the points raised during the review process.

Specifically, you must provide assurance/make plans to deposit your fungal strains in a collection outside China, so that they are accessible to laboratories world-wide.  This is a requirement of PLOS journals. 

We look forward to receiving your revised manuscript.

Kind regards,

Katherine A. Borkovich, Ph.D.

Academic Editor

PLOS ONE

Journal Requirements:

1. PLOS ONE does not provide copyediting or proofs of accepted manuscripts, we therefore  recommend that you carefully review your manuscript and correct any spelling errors as suggested by Reviewer 2.

Additional Editor Comments (if provided):

Reviewers' comments:

Reviewer's Responses to Questions

**Comments to the Author**

1. If the authors have adequately addressed your comments raised in a previous round of review and you feel that this manuscript is now acceptable for publication, you may indicate that here to bypass the “Comments to the Author” section, enter your conflict of interest statement in the “Confidential to Editor” section, and submit your "Accept" recommendation.

Reviewer #1: All comments have been addressed

Reviewer #2: (No Response)

Reviewer #3: All comments have been addressed

2. Is the manuscript technically sound, and do the data support the conclusions?

Reviewer #1: (No Response)

Reviewer #2: Yes

Reviewer #3: Yes

3. Has the statistical analysis been performed appropriately and rigorously? 

Reviewer #1: (No Response)

Reviewer #2: N/A

Reviewer #3: Yes

4. Have the authors made all data underlying the findings in their manuscript fully available?

Reviewer #1: (No Response)

Reviewer #2: No

Reviewer #3: Yes

5. Is the manuscript presented in an intelligible fashion and written in standard English?

Reviewer #1: (No Response)

Reviewer #2: Yes

Reviewer #3: Yes

6. Review Comments to the Author

Reviewer #1: (No Response)

Reviewer #2: The reviewer’s response to the rebuttal by the authors is indicated by a "#" in front. With the specific comments, the "#"-marked-up response is placed directly below the authors’ replies.

#General comments of the reviewer: The reviewer thinks that many of his previous comments have been addressed adequately. The most important exception from this is the still unsatisfying (promise on) availability of the employed F. filiformis strains: The reference by the authors of the deposition of F. filiformis M1 (CCMSSC04554) in the mentioned Chinese culture collection is definitely not enough to ensure availability of the strain for researchers outside China to be able to assess this biological material independently of collaboration with Chinese researchers, for the previously mentioned reason of a restrictive Chinese governmental policy on national resources. In addition, the reviewer assumes that just the promise of depositing both strains of their study (F. filiformis M1 and F. filiformis XR) in a fungal strain collection outside China after the COVID-19 pandemic crisis is settled (when? in 2021?), is certainly not enough to satisfy the PLoS ONE requirements of making research data and material available before the manuscript is accepted for publication.

As a compromise, if deposition of the strains may be easier to conduct (less formalities etc.) at a non-Chinese research institute/lab also working on Flammulina, e.g. the van Peer lab (WUR, Wageningen, NL), such may also be an acceptable minimalist way to ensure that colleagues outside China can access the strains independently. The principal investigator of the van Peer lab has worked on Flammulina in China, so he may be considered (re)liable enough by Chinese authorities.

Until the strain availability issue is dealt with more convincingly, for now, this reviewer refrains from recommending endorsement of the manuscript for publication.

Specific comments:

- Abstract: The authors speak of >200 reference genomes which were analysed to fish for conserved genes playing a role in mushroom formation. However, they seem not to explain this in detail further down in the manuscript, e.g. in the Material and methods section.

Reply: Thanks for your suggestion. We add it in Material and methods section in Line

203-209.

#Reviewer’s comment: Thanks for adding this information now.

- Introduction:

o Attention is paid to the two traditional agaric model systems Schizophyllum commune and Coprinopsis cinerea in which advances in molecular genetics techniques allowed to study of fruiting body formation relevant genes. In addition, the authors speak of economically important non-model systems in scientific focus since more recently but do not cite the worldwide cultivated edible mushroom Cyclocybe aegerita (poplar mushroom; formerly Agrocybe aegerita) which in this context, besides its economic importance, seems not only interesting enough to mention for its capability of haploid fruiting without mating and bioactive compound production but, since recently, also allows functional genetics analyses (MGG 294:663-677. https://doi.org/10.1007/s00438-018-01528-6; AEM 85:e01549-19. https://doi.org/10.1128/AEM.01549-19; Beilstein J. Org. Chem. 2019, 15, 1000–1007. doi:10.3762/bjoc.15.98).

Reply: Thank you very much. We cite this work in introduction section. Line88-91.

#Reviewer’s comment: Although the authors now mention the fungus, they include a typo when writing its genus name: it must be spelled Cyclocybe aegerita (not Cyclogybe!). In addition, the reviewer notices that the authors only cite the work on the recently established genetic system of this rather important fungal species, but they leave out the equally interesting work on the first Basidiomycota ribotoxin Ageritin published with this species (AEM 85:e01549-19. https://doi.org/10.1128/AEM.01549-19). The reviewer kindly asks the authors to also cite this one not least given the potential wide distribution of this basidiomycete ribotoxin among diverse agaricomycetes (including Pleurotus species where pore-forming defense proteins to which category the authors suggest flammutoxin to also belong to), at least when discussing their flammutoxin further down in context of other mushroom toxins (see recommendations below).

By the way: the reviewer also noticed another typo in line 85 when the authors speak about CRISPR/Cas9-methodology established with C. cinerea and S. commune. It must not write “CRISPER/Cas9”…

- Material & Methods:

o The two investigated strains of F. filiformis do not seem to be available from an internationally accessible strain collection like the Westerdijk Institute (Utrecht, Netherlands) or the ATCC (American Type Culture Collection in Manassas, Virginia, United States). How will the authors guarantee that scientists outside China will not be refused official requests for F. filiformis M1 (CCMSSC04554) and F. filiformis XR for research purposes (independent from collaboration with Chinese scientists) in view of the very restrictive Chinese policy on national bioresources normally resulting in refusal of biomaterial issuance to scientists outside China?

Reply: We are very willing to deposit our strain in relevant collection center. Indeed, one of the strains (F. filiformis M1, CCMSSC04554) in our study already deposited in China Center for Mushroom Spawn Standards and Control. We would like to deposit them in other international collection center after the pandemic of COVID-19.

#Reviewer’s comment: As said above, the reviewer assumes that just the promise of depositing both strains of their study (F. filiformis M1 and F. filiformis XR) in a fungal strain collection outside China after the COVID-19 pandemic crisis is settled (when? in 2021?), is certainly not enough to satisfy the PLoS ONE requirements of making research data and material available before the manuscript is accepted for publication suggesting a potential compromise.

o Furthermore, the reviewer thinks that details should be given on the exact geographic and, if available, the habitat origin of both strains (something like “…strain XR was isolated from a mushroom market…”, line 126, is hardly informative).

Reply: Done. Line136-139.

#Reviewer’s comment: Thank you very much!

o Apart from that the reviewer suggests to correct every mentioning of fungal strains in this manuscript so far to always mention them together with the species name, e.g. F. filiformis M1 and F. filiformis XR.

#Reviewer’s comment: This previous comment of mine was not addressed in the authors’ point-by-point rebuttal and also not in the revised manuscript!

o The reviewer sees one more shortcoming in this section (RNA-sampling and –sequencing was carried out accurately): the authors should specifically mention how they performed the realtime quantitative PCR (qRT-PCR) assessment of the differential expression of the 22 randomly selected fruiting- and heat-stress correlated genes they mention further down (Fig. S1). This is to include in a small paragraph on qRT-PCR in this section before the study can be accepted for publication in PLoS One.

Reply: Done, we add it in materials and methods section, in Line210-Lin219.

#Reviewer’s comment: Thanks for adding this. Still, please take care to always leave a space between numbers and unit (also throughout in the manuscript => please recheck this everywhere) such as in line 214: “10min”, “5µL” etc. => please correct to: “10 min”, 25 µL” etc.; In addition, please everywhere give “µL” not “uL” as in line 215! Furthermore, a line break in line 220 transferring “Results and Discussion” to the next page would be useful.

In addition, the figure caption of Fig. S1 should be more informative (the authors should double-check on this also in their other figures): what will “chr11_AA_01521” or “P10-M1” tell the uninformed reader of the study? The reviewer is aware that fruiting stage describing abbreviations like “P10-M1” are spelled out further down in the figure legend of Fig. 2 but explaining these terms in in Fig. S1, too seems adequate to increase readability of the manuscript. Apart from that, there seems to be a discrepancy between the 22 genes checked with qRT-PCR in the manuscript text (line 201) and in the figure legend where it states 18 genes were qRT-PCR-checked (line 695)!

Reply: We add the legend in Fig. S1, the number of the genes involved in qRT-PCR should be 18. Thank you very much.

#Reviewer’s comment: Thanks for correcting the discrepancy and adding the what the abbreviations mean in the figure legend of Fig. S1.

Terms like “chr11_...” and chr05…” also imply that the reads were potentially mapped to chromosome level of the F. filiformis genome. Could this please be specified somewhere in this section or elsewhere appropriate in the manuscript?

Reply: Yes, the reference genome was assembled in chromosome level, we mentioned it in Line176.

#Reviewer’s comment: Thank you very much.

- Results and Discussion:

o Line 268 and Tab. 1: It is mentioned that the flammutoxin-encoding gene is conserved (if this is a typo there "...conversed...", line 268; it should also be "...was..." instead of "...were..." I assume) in 20 Basidiomycota species.

Reply: Done. Thank you very much.

#Reviewer’s comment: Thank you very much.

It seems interesting to compare whether other anti-antagonist mushroom compounds like galectins, ribotoxins, or other antimicrobial proteins/peptides are also conserved and expressed (When? In vegetative mycelium or during fruiting?) in F. filiformis? The authors should check on this as well and possibly complement their results with this recommendable checkup.

Reply: Thank you very much for this suggestion. We didn’t find the galectins and

ribotoxins coding gene in F. filiformis genome. Whereas, we find Macrolepiota procera mpl homologous gene in F. filiformis. This gene is conserved in Agaricomycetes, and have a conserved expression pattern in fruiting body stage. We discussed its role in line285-lin292.

#Reviewer’s comment: Thanks for adding this discussion piece on the Mpl protein of M. procera. I have two friendly calls for correction/modification here: first, the reviewer thinks that the finding by the authors that the F. filiformis genome apparently not contains other relevant mushroom defense proteins like galectin- and ribotoxin-encoding genes is interesting enough to explicitly mention it in the manuscript text shortly (within lines 285 to 292), not least since, for instance, the above mentioned study of Tayyrov et al. (2019; AEM 85:e01549-19) shows that potential basidiomycete ribotoxin orthologs should be quite widely spread among very different agaricomycetes (e.g., among different agarics and boletes, including Pleurotus spp. where, interestingly, pore-forming defense proteins like flammutoxin, assumingly, should also be present in addition to putative ageritin-like ribotoxin genes). Second, there is a typo in line 287: “bodyies” must be changed to “bodies”.

o Lines 278-280: Maybe the authors should like to shortly emphasise the need for a resilient functional genetics toolbox in F. filiformis that would greatly serve increasing knowledge about this interesting gene? In this context, they may like to discuss which tools (transformation techniques, transformant selection markers, overexpression tools,...) are available for F. filiformis.

Reply: Done, Line296-Line298. Thank you very much.

#Reviewer’s comment: Thank you very much for adding this.

o One general comment on gene notation: Either the authors follow the widely-applied “basidiomycete code” (wild type gene names written like in yeast code but all letters lowercase; protein names as in yeast code…) or the “yeast code” (wild type gene names written italicised in uppercase letters with a number; proteins non-italicised, first letter uppercase; this code is sometimes applied in Basidiomycota, too, especially when addressing gene names of a certain species like AaeAGT1 for the ribotoxin gene of the poplar mushroom to not get somewhat “inartfully” looking gene names like Aaeagt1). This should be corrected before publication.

Reply: Done. Thank you very much.

#Reviewer’s comment: Thanks for correcting this.

Reviewer #3: This study examined mushroom forming fungi, with an emphasis on how the winter mushroom, Flammulina filiformis, development responds to heat stress. It was an interesting study, and would be relevant for mushroom cultivators, and also for understanding how we may use transcriptomics and comparative genomics to identify conserved genes that may play important roles in fruiting body development and heat stress. Although only a portion of my comments were incorporated into the draft, the authors considered my suggestions. Furthermore, I believe that the authors have substantively improved the draft, since the previous version. Therefore I recommend this manuscript to be accepted for publication.

7. PLOS authors have the option to publish the peer review history of their article (what does this mean?). If published, this will include your full peer review and any attached files.

Reviewer #1: No

Reviewer #2: No

Reviewer #3: No

---

## [Author Response · Author response to Decision Letter 1]

15 Jul 2020

Dear Editors of PLoS One,

Thank you very much for your email with regard to our manuscript entitled “Transcriptome data reveal conserved patterns of fruiting body development and response to heat stress in the mushroom-forming fungus Flammulina filiformis” (PONE-D-20-02460). We carefully revised the critiques, comments and suggestions from the reviewers. 

For the strain deposit, we have getting in touch with Westerdijk Fungal Biodiversity Institute, CBS Collection, and got agreement to deposit the two strains (M1 and XR) in CBS. Thus, we posted them by FedEx on 10th July (Track ID: 813740466856). Unfraternally, they were forbidden to export by Customs of China. Therefore, we would like to post them again after the pandemic of COVID-19.

We really appreciate your kind help. We believe that the revised manuscript is much improved and hope it will be acceptable for publication in PLoS One.

We look forward to your further communications with regard to this manuscript. Below are our responses to the comments raises by the reviewers.

Best wishes,

Xiao-Bin Liu, László G. Nagy, Zhu L. Yang

Reviewer #2: 

#General comments of the reviewer: The reviewer thinks that many of his previous comments have been addressed adequately. The most important exception from this is the still unsatisfying (promise on) availability of the employed F. filiformis strains: The reference by the authors of the deposition of F. filiformis M1 (CCMSSC04554) in the mentioned Chinese culture collection is definitely not enough to ensure availability of the strain for researchers outside China to be able to assess this biological material independently of collaboration with Chinese researchers, for the previously mentioned reason of a restrictive Chinese governmental policy on national resources. In addition, the reviewer assumes that just the promise of depositing both strains of their study (F. filiformis M1 and F. filiformis XR) in a fungal strain collection outside China after the COVID-19 pandemic crisis is settled (when? in 2021?), is certainly not enough to satisfy the PLoS ONE requirements of making research data and material available before the manuscript is accepted for publication.

As a compromise, if deposition of the strains may be easier to conduct (less formalities etc.) at a non-Chinese research institute/lab also working on Flammulina, e.g. the van Peer lab (WUR, Wageningen, NL), such may also be an acceptable minimalist way to ensure that colleagues outside China can access the strains independently. The principal investigator of the van Peer lab has worked on Flammulina in China, so he may be considered (re)liable enough by Chinese authorities.

Until the strain availability issue is dealt with more convincingly, for now, this reviewer refrains from recommending endorsement of the manuscript for publication.

Reply: We have getting in touch with Westerdijk Fungal Biodiversity Institute, CBS Collection, and got agreement to deposit the two strains (M1 and XR) in CBS. Thus, we post them by FedEx on 10th July (Track ID: 813740466856). Unfraternally, they were forbidden to export by Customs of China. Therefore, we would like to post them again after the pandemic of COVID-19.

Specific comments:

- Introduction:

o Attention is paid to the two traditional agaric model systems Schizophyllum commune and Coprinopsis cinerea in which advances in molecular genetics techniques allowed to study of fruiting body formation relevant genes. In addition, the authors speak of economically important non-model systems in scientific focus since more recently but do not cite the worldwide cultivated edible mushroom Cyclocybe aegerita (poplar mushroom; formerly Agrocybe aegerita) which in this context, besides its economic importance, seems not only interesting enough to mention for its capability of haploid fruiting without mating and bioactive compound production but, since recently, also allows functional genetics analyses (MGG 294:663-677. https://doi.org/10.1007/s00438-018-01528-6; AEM 85:e01549-19. https://doi.org/10.1128/AEM.01549-19; Beilstein J. Org. Chem. 2019, 15, 1000–1007. doi:10.3762/bjoc.15.98).

Reply: Thank you very much. We cite this work in introduction section. Line88-91.

#Reviewer’s comment: Although the authors now mention the fungus, they include a typo when writing its genus name: it must be spelled Cyclocybe aegerita (not Cyclogybe!). In addition, the reviewer notices that the authors only cite the work on the recently established genetic system of this rather important fungal species, but they leave out the equally interesting work on the first Basidiomycota ribotoxin Ageritin published with this species (AEM 85:e01549-19. https://doi.org/10.1128/AEM.01549-19). The reviewer kindly asks the authors to also cite this one not least given the potential wide distribution of this basidiomycete ribotoxin among diverse agaricomycetes (including Pleurotus species where pore-forming defense proteins to which category the authors suggest flammutoxin to also belong to), at least when discussing their flammutoxin further down in context of other mushroom toxins (see recommendations below).

By the way: the reviewer also noticed another typo in line 85 when the authors speak about CRISPR/Cas9-methodology established with C. cinerea and S. commune. It must not write “CRISPER/Cas9”…

Reply: Thanks very much, we revised Cyclogybe to Cyclocybe, and CRISPER/Cas9 to CRISPR/Cas9. We cited Tayyrov et al. 2019 in line 294-296.

- Material & Methods:

o The two investigated strains of F. filiformis do not seem to be available from an internationally accessible strain collection like the Westerdijk Institute (Utrecht, Netherlands) or the ATCC (American Type Culture Collection in Manassas, Virginia, United States). How will the authors guarantee that scientists outside China will not be refused official requests for F. filiformis M1 (CCMSSC04554) and F. filiformis XR for research purposes (independent from collaboration with Chinese scientists) in view of the very restrictive Chinese policy on national bioresources normally resulting in refusal of biomaterial issuance to scientists outside China?

Reply: We are very willing to deposit our strain in relevant collection center. Indeed, one of the strains (F. filiformis M1, CCMSSC04554) in our study already deposited in China Center for Mushroom Spawn Standards and Control. We would like to deposit them in other international collection center after the pandemic of COVID-19.

#Reviewer’s comment: As said above, the reviewer assumes that just the promise of depositing both strains of their study (F. filiformis M1 and F. filiformis XR) in a fungal strain collection outside China after the COVID-19 pandemic crisis is settled (when? in 2021?), is certainly not enough to satisfy the PLoS ONE requirements of making research data and material available before the manuscript is accepted for publication suggesting a potential compromise.

#Reviewer’s comment: This previous comment of mine was not addressed in the authors’ point-by-point rebuttal and also not in the revised manuscript!

Reply: Since we mentioned from the title of which this study is focused on the species F. filiformis, and we also mentioning the strain M1 and XR belongs to F. filiformis in abstract. Therefore, in order to keep the manuscript concise, we didn’t add the species name F. filiformis together with the strain name in other parts of this manuscript.

o The reviewer sees one more shortcoming in this section (RNA-sampling and –sequencing was carried out accurately): the authors should specifically mention how they performed the realtime quantitative PCR (qRT-PCR) assessment of the differential expression of the 22 randomly selected fruiting- and heat-stress correlated genes they mention further down (Fig. S1). This is to include in a small paragraph on qRT-PCR in this section before the study can be accepted for publication in PLoS One.

Reply: Done, we add it in materials and methods section, in Line210-Lin219.

#Reviewer’s comment: Thanks for adding this. Still, please take care to always leave a space between numbers and unit (also throughout in the manuscript => please recheck this everywhere) such as in line 214: “10min”, “5µL” etc. => please correct to: “10 min”, 25 µL” etc.; In addition, please everywhere give “µL” not “uL” as in line 215! Furthermore, a line break in line 220 transferring “Results and Discussion” to the next page would be useful

Reply: Thank you very much, done.

- Results and Discussion:

It seems interesting to compare whether other anti-antagonist mushroom compounds like galectins, ribotoxins, or other antimicrobial proteins/peptides are also conserved and expressed (When? In vegetative mycelium or during fruiting?) in F. filiformis? The authors should check on this as well and possibly complement their results with this recommendable checkup.

Reply: Thank you very much for this suggestion. We didn’t find the galectins and

ribotoxins coding gene in F. filiformis genome. Whereas, we find Macrolepiota procera mpl homologous gene in F. filiformis. This gene is conserved in Agaricomycetes, and have a conserved expression pattern in fruiting body stage. We discussed its role in line285-lin292.

#Reviewer’s comment: Thanks for adding this discussion piece on the Mpl protein of M. procera. I have two friendly calls for correction/modification here: first, the reviewer thinks that the finding by the authors that the F. filiformis genome apparently not contains other relevant mushroom defense proteins like galectin- and ribotoxin-encoding genes is interesting enough to explicitly mention it in the manuscript text shortly (within lines 285 to 292), not least since, for instance, the above mentioned study of Tayyrov et al. (2019; AEM 85:e01549-19) shows that potential basidiomycete ribotoxin orthologs should be quite widely spread among very different agaricomycetes (e.g., among different agarics and boletes, including Pleurotus spp. where, interestingly, pore-forming defense proteins like flammutoxin, assumingly, should also be present in addition to putative ageritin-like ribotoxin genes). Second, there is a typo in line 287: “bodyies” must be changed to “bodies”.

Reply: Thanks for this suggestion. We add the short comments about the ribotoxin in line 294-296. For the typo of “bodyies”, we revised it.

---

## [Editor Report · Decision Letter 2]

10 Aug 2020

PONE-D-20-02460R2

Transcriptome data reveal conserved patterns of fruiting body development and response to heat stress in the mushroom-forming fungus Flammulina filiformis

PLOS ONE

Dear Dr. Liu,

I apologize for the delay in getting back to you about your manuscript. Because your fungal strains are part of your minimal data set (which is used to reach the conclusions drawn in the manuscript with related metadata and methods, and any additional data required to replicate the reported study findings in their entirety), they must be publicly available before publication.

I understand that you are unable to make these strains available at this time. Therefore, I have been advised by the journal to render a "Revise" decision to give you time to eventually deposit them and provide the deposit information in your final manuscript.

We look forward to receiving your revised manuscript.

Kind regards,

Katherine A. Borkovich, Ph.D.

Academic Editor

PLOS ONE

---

## [Author Response · Author response to Decision Letter 2]

15 Sep 2020

Dear Editors of PLoS One,

Thank you very much for your email with regard to our manuscript entitled “Transcriptome data reveal conserved patterns of fruiting body development and response to heat stress in the mushroom-forming fungus Flammulina filiformis” (PONE-D-20-02460). We carefully revised the critiques, comments and suggestions from the reviewers. 

For the strain deposit, we deposit our strain in “The Chinese General Microbiological Culture Collection Center (CGMCC)”. Strains deposit in this center are officially announced available to the international scientific community in both academic and industrial institutions (http://www.cgmcc.net/english/Deposit.html). We add the strain number in our manuscript in Materials and methods in Line: 135-139.

We really appreciate your kind help. We believe that the revised manuscript is much improved and hope it will be acceptable for publication in PLoS One.

We look forward to your further communications with regard to this manuscript. Below are our responses to the comments raises by the reviewers.

Best wishes,

Xiao-Bin Liu, László G. Nagy, Zhu L. Yang

Reviewer #2: 

#General comments of the reviewer: The reviewer thinks that many of his previous comments have been addressed adequately. The most important exception from this is the still unsatisfying (promise on) availability of the employed F. filiformis strains: The reference by the authors of the deposition of F. filiformis M1 (CCMSSC04554) in the mentioned Chinese culture collection is definitely not enough to ensure availability of the strain for researchers outside China to be able to assess this biological material independently of collaboration with Chinese researchers, for the previously mentioned reason of a restrictive Chinese governmental policy on national resources. In addition, the reviewer assumes that just the promise of depositing both strains of their study (F. filiformis M1 and F. filiformis XR) in a fungal strain collection outside China after the COVID-19 pandemic crisis is settled (when? in 2021?), is certainly not enough to satisfy the PLoS ONE requirements of making research data and material available before the manuscript is accepted for publication.

As a compromise, if deposition of the strains may be easier to conduct (less formalities etc.) at a non-Chinese research institute/lab also working on Flammulina, e.g. the van Peer lab (WUR, Wageningen, NL), such may also be an acceptable minimalist way to ensure that colleagues outside China can access the strains independently. The principal investigator of the van Peer lab has worked on Flammulina in China, so he may be considered (re)liable enough by Chinese authorities.

Until the strain availability issue is dealt with more convincingly, for now, this reviewer refrains from recommending endorsement of the manuscript for publication.

Reply: We have getting in touch with Westerdijk Fungal Biodiversity Institute, CBS Collection, and got agreement to deposit the two strains (M1 and XR) in CBS. Thus, we post them by FedEx on 10th July (Track ID: 813740466856). Unfraternally, they were forbidden to export by Customs of China. Afterwards, we deposit our strain in “The Chinese General Microbiological Culture Collection Center (CGMCC)”. Strains deposit in this center are officially announced available to the international scientific community in both academic and industrial institutions (http://www.cgmcc.net/english/Deposit.html). We add the strain number in our manuscript in Materials and methods in Line: 135-139.

Specific comments:

- Introduction:

o Attention is paid to the two traditional agaric model systems Schizophyllum commune and Coprinopsis cinerea in which advances in molecular genetics techniques allowed to study of fruiting body formation relevant genes. In addition, the authors speak of economically important non-model systems in scientific focus since more recently but do not cite the worldwide cultivated edible mushroom Cyclocybe aegerita (poplar mushroom; formerly Agrocybe aegerita) which in this context, besides its economic importance, seems not only interesting enough to mention for its capability of haploid fruiting without mating and bioactive compound production but, since recently, also allows functional genetics analyses (MGG 294:663-677. https://doi.org/10.1007/s00438-018-01528-6; AEM 85:e01549-19. https://doi.org/10.1128/AEM.01549-19; Beilstein J. Org. Chem. 2019, 15, 1000–1007. doi:10.3762/bjoc.15.98).

Reply: Thank you very much. We cite this work in introduction section. Line88-91.

#Reviewer’s comment: Although the authors now mention the fungus, they include a typo when writing its genus name: it must be spelled Cyclocybe aegerita (not Cyclogybe!). In addition, the reviewer notices that the authors only cite the work on the recently established genetic system of this rather important fungal species, but they leave out the equally interesting work on the first Basidiomycota ribotoxin Ageritin published with this species (AEM 85:e01549-19. https://doi.org/10.1128/AEM.01549-19). The reviewer kindly asks the authors to also cite this one not least given the potential wide distribution of this basidiomycete ribotoxin among diverse agaricomycetes (including Pleurotus species where pore-forming defense proteins to which category the authors suggest flammutoxin to also belong to), at least when discussing their flammutoxin further down in context of other mushroom toxins (see recommendations below).

By the way: the reviewer also noticed another typo in line 85 when the authors speak about CRISPR/Cas9-methodology established with C. cinerea and S. commune. It must not write “CRISPER/Cas9”…

Reply: Thanks very much, we revised Cyclogybe to Cyclocybe, and CRISPER/Cas9 to CRISPR/Cas9. We cited Tayyrov et al. 2019 in line 294-296.

- Material & Methods:

o The two investigated strains of F. filiformis do not seem to be available from an internationally accessible strain collection like the Westerdijk Institute (Utrecht, Netherlands) or the ATCC (American Type Culture Collection in Manassas, Virginia, United States). How will the authors guarantee that scientists outside China will not be refused official requests for F. filiformis M1 (CCMSSC04554) and F. filiformis XR for research purposes (independent from collaboration with Chinese scientists) in view of the very restrictive Chinese policy on national bioresources normally resulting in refusal of biomaterial issuance to scientists outside China?

Reply: We are very willing to deposit our strain in relevant collection center. Indeed, one of the strains (F. filiformis M1, CCMSSC04554) in our study already deposited in China Center for Mushroom Spawn Standards and Control. We would like to deposit them in other international collection center after the pandemic of COVID-19.

#Reviewer’s comment: As said above, the reviewer assumes that just the promise of depositing both strains of their study (F. filiformis M1 and F. filiformis XR) in a fungal strain collection outside China after the COVID-19 pandemic crisis is settled (when? in 2021?), is certainly not enough to satisfy the PLoS ONE requirements of making research data and material available before the manuscript is accepted for publication suggesting a potential compromise.

Reply: We have getting in touch with Westerdijk Fungal Biodiversity Institute, CBS Collection, and got agreement to deposit the two strains (M1 and XR) in CBS. Thus, we post them by FedEx on 10th July (Track ID: 813740466856). Unfraternally, they were forbidden to export by Customs of China. Afterwards, we deposit our strain in “The Chinese General Microbiological Culture Collection Center (CGMCC)”. Strains deposit in this center are officially announced available to the international scientific community in both academic and industrial institutions (http://www.cgmcc.net/english/Deposit.html). We add the strain number in our manuscript in Materials and methods in Line: 135-139.

#Reviewer’s comment: This previous comment of mine was not addressed in the authors’ point-by-point rebuttal and also not in the revised manuscript!

Reply: Since we mentioned from the title of which this study is focused on the species F. filiformis, and we also mentioning the strain M1 and XR belongs to F. filiformis in abstract. Therefore, in order to keep the manuscript concise, we didn’t add the species name F. filiformis together with the strain name in other parts of this manuscript.

o The reviewer sees one more shortcoming in this section (RNA-sampling and –sequencing was carried out accurately): the authors should specifically mention how they performed the realtime quantitative PCR (qRT-PCR) assessment of the differential expression of the 22 randomly selected fruiting- and heat-stress correlated genes they mention further down (Fig. S1). This is to include in a small paragraph on qRT-PCR in this section before the study can be accepted for publication in PLoS One.

Reply: Done, we add it in materials and methods section, in Line210-Lin219.

#Reviewer’s comment: Thanks for adding this. Still, please take care to always leave a space between numbers and unit (also throughout in the manuscript => please recheck this everywhere) such as in line 214: “10min”, “5µL” etc. => please correct to: “10 min”, 25 µL” etc.; In addition, please everywhere give “µL” not “uL” as in line 215! Furthermore, a line break in line 220 transferring “Results and Discussion” to the next page would be useful

Reply: Thank you very much, done.

- Results and Discussion:

It seems interesting to compare whether other anti-antagonist mushroom compounds like galectins, ribotoxins, or other antimicrobial proteins/peptides are also conserved and expressed (When? In vegetative mycelium or during fruiting?) in F. filiformis? The authors should check on this as well and possibly complement their results with this recommendable checkup.

Reply: Thank you very much for this suggestion. We didn’t find the galectins and

ribotoxins coding gene in F. filiformis genome. Whereas, we find Macrolepiota procera mpl homologous gene in F. filiformis. This gene is conserved in Agaricomycetes, and have a conserved expression pattern in fruiting body stage. We discussed its role in line285-lin292.

#Reviewer’s comment: Thanks for adding this discussion piece on the Mpl protein of M. procera. I have two friendly calls for correction/modification here: first, the reviewer thinks that the finding by the authors that the F. filiformis genome apparently not contains other relevant mushroom defense proteins like galectin- and ribotoxin-encoding genes is interesting enough to explicitly mention it in the manuscript text shortly (within lines 285 to 292), not least since, for instance, the above mentioned study of Tayyrov et al. (2019; AEM 85:e01549-19) shows that potential basidiomycete ribotoxin orthologs should be quite widely spread among very different agaricomycetes (e.g., among different agarics and boletes, including Pleurotus spp. where, interestingly, pore-forming defense proteins like flammutoxin, assumingly, should also be present in addition to putative ageritin-like ribotoxin genes). Second, there is a typo in line 287: “bodyies” must be changed to “bodies”.

Reply: Thanks for this suggestion. We add the short comments about the ribotoxin in line 294-296. For the typo of “bodyies”, we revised it.

---

## [Editor Report · Decision Letter 3]

16 Sep 2020

Transcriptome data reveal conserved patterns of fruiting body development and response to heat stress in the mushroom-forming fungus Flammulina filiformis

PONE-D-20-02460R3

Dear Dr. Liu,

We’re pleased to inform you that your manuscript has been judged scientifically suitable for publication and will be formally accepted for publication once it meets all outstanding technical requirements. Considering we are still operating under the COVID pandemic, with limited ability to ship materials internationally in many cases, I am accepting your deposition of the strains in the Chinese General Microbiological Culture Collection Center as sufficient to satisfy the requirement for making materials freely available for acceptance of your paper.  

Kind regards,

Katherine A. Borkovich, Ph.D.

Academic Editor

PLOS ONE
---

## [Editor Report · Acceptance letter]

6 Oct 2020

PONE-D-20-02460R3 

Transcriptome data reveal conserved patterns of fruiting body development and response to heat stress in the mushroom-forming fungus *Flammulina filiformis*

Dear Dr. Liu:

I'm pleased to inform you that your manuscript has been deemed suitable for publication in PLOS ONE. Congratulations! Your manuscript is now with our production department. 

Kind regards, 

on behalf of

Dr. Katherine A. Borkovich 

Academic Editor

PLOS ONE